# Transported African Dust in the Lower Marine Atmospheric Boundary Layer is Internally Mixed with Sea Salt Contributing to Increased Hygroscopicity and a Lower Lidar Depolarization Ratio

Sujan Shrestha[1], Robert E. Holz[2*], Willem J. Marais[2], Zachary Buckholtz[2], Ilya Razenkov[2], Edwin Eloranta[2], Jeffrey S. Reid[3], Hope E. Elliott[1,a], Nurun Nahar Lata[4], Zezhen Cheng[4], Swarup China[4], Edmund Blades[1], Albert D. Ortiz[1], Rebecca Chewitt-Lucas[5], Alyson Allen[1], Devon Blades[1], Ria Agrawal[1], Elizabeth A. Reid[3], Jesus Ruiz-Plancarte[6], Anthony Bucholtz[6], Ryan Yamaguchi[6], Qing Wang[6], Thomas Eck[7], Elena Lind[7], Mira L. Pöhlker[8], Andrew P. Ault[9], Cassandra J. Gaston[1*]

[1]Department of Atmospheric Sciences, Rosenstiel School of Marine, Atmospheric, and Earth Science, University of Miami, FL, USA

[2]Space Science and Engineering Center (SSEC), University of Wisconsin-Madison, WI, USA

[3]U.S. Naval Research Laboratory, Monterey, CA, USA

[4]Environmental Molecular Sciences Laboratory, Pacific Northwest National Laboratory, Richland, WA, USA

[5]Caribbean Institute for Meteorology and Hydrology, Barbados

[6]Department of Meteorology, Naval Postgraduate School, Monterey, CA, USA

[7]NASA Goddard Space Flight Center, Greenbelt, MD, USA

[8]Atmospheric Microphysics Department, Leibniz Institute for Tropospheric Research, Leipzig, Germany

[9]Department of Chemistry, University of Michigan, MI, USA

[a]now at Department of Biological & Environmental Sciences, Wittenberg University, Springfield, OH, USA

*Corresponding Author: Cassandra J. Gaston: Email: cgaston@miami.edu, Phone: (305)-421-4979 and Robert E. Holz: Email: reholz@ssec.wisc.edu

**Abstract**

27  Saharan dust is frequently transported across the Atlantic, yet the chemical, physical, and

morphological transformations dust undergoes within the marine atmospheric boundary layer
(MABL) remain poorly understood. These transformations are critical for understanding dust's
radiative and geochemical impacts, it's representation in atmospheric models, and detection via
remote sensing. Here, we present coordinated observations from the Office of Naval Research's
Moisture and Aerosol Gradients/Physics of Inversion Evolution (MAGPIE) August 2023
campaign at Ragged Point, Barbados. These include vertically resolved single-particle analyses,
mass concentrations of dust and sea spray, and High Spectral Resolution Lidar (HSRL)
retrievals. Single-particle data show that dust within the Saharan Air Layer (SAL) remains
externally mixed, with a corresponding high HSRL-derived linear depolarization ratio (LDR) at
532 nm of ~0.3. However, at lower altitudes, dust becomes internally mixed with sea spray, and
under the high humidity (>80%) of the MABL undergoes hygroscopic growth, yielding more
spherical particles, suppressing the LDR to <0.1; even in the presence of high dust loadings (e.g.,
~120 $\mu g/m^3$). This low depolarization in the MABL is likely due to a combination of the
differences between the single scattering properties of dust and spherical particles, and the
potential modification of the dust optical properties from an increased hygroscopicity of dust
caused by the mixing with sea salt in the humid MABL. These results highlight the importance
of the aerosol particle mixing state when interpreting LDR-derived dust retrievals and estimating
surface dust concentrations in satellite products and atmospheric models.

## 1. Introduction

The transport of Saharan dust across the North Atlantic Basin throughout the year is one of the largest aerosol phenomena observable from space. The most intensive events often occur during the boreal summer when large quantities of dust are lofted and advected westward by trade winds within the Saharan Air Layer (SAL), a well-defined elevated layer extending from ~2 to 5 km above mean sea level (AMSL) (e.g., Carlson and Prospero, 1972; Karyampudi et al., 1999; Adams et al., 2012; Tsamalis et al., 2013; Mehra et al., 2023). This conceptual model of African dust transport is frequently reinforced by satellite and ground-based remote sensing, particularly lidar (Burton et al., 2012, 2015), multi-angle imager (Kalashnikova et al., 2013), polarimetric (Huang et al., 2015) or combination of these observations (Moustaka et al., 2025) that rely on dust's asphericity to differentiate coarse mode dust from other aerosol sources such as hydrated sea spray. These techniques often detect little dust within the lower marine atmospheric boundary layer (MABL). However, it is well known that exceptionally high dust concentrations are often directly measured in the MABL (e.g., Reid et al., 2003b; Zuidema et al., 2019; Elliott et al., 2024; Mayol-Bracero et al., 2025) and these layers are regularly forecast by operational dust transport models (Xian et al., 2019). This contradiction between the common conceptual model fueled by remote sensing of elevated dust layers versus evidence of significant near-surface dust mass concentrations by in situ observations raises a critical question, is there an observational gap in the detection and characterization of dust within the MABL?

Among the methods to speciate airborne dust from other aerosol particle types, the most common benchmark is to rely on dust's asphericity, and its impact on lidar's linear depolarization ratio (LDR). The LDR is based on a lidar's range-resolved measurement of the fraction of backscattered light by aerosol particles that become depolarized from the original polarized laser

pulse. Backscattered light from homogeneous spherical particles, such as hydrated sea salt, has
low depolarization (e.g., LDR remains minimal) whereas particles with asymmetry such as dry,
irregular dust will return a partially depolarized signal, typically ~0.25-0.40 (Murayama et al.,
1999; Ansmann et al., 2012; Burton et al., 2012; Freudenthaler et al., 2009; Sakai et al., 2010;
Groß et al., 2016).

The assertion that dust can be isolated from other aerosol types such as in the references

above is well supported by both theoretical foundations and numerous observations of elevated
dust plumes. An important assumption in the detection of dust via the LDR is that the dust is not
hygroscopic. In situ observations of dust hygroscopicity in the MABL, typically using the
standard technique of drying and subsequently rehydrating particles ahead of nephelometer
measurements (Orozco et al., 2016), have suggested MABL dust is not significantly hygroscopic
(Li-Jones et al., 1998; Zhang et al., 2014). Thus, it is often assumed that dust in the humid
MABL will retain its aspherical shape and remain tracible via the LDR. However, even freshly
emitted dust or that which is sampled well within a dust plume can contain soluble minerals that
should be inherently hygroscopic and could affect detection of dust via the LDR (Koehler et al.,
2007; Reid et al., 2003a).

Contradictory observations have introduced uncertainty in the interpretation of lidar

observations for dust detection in the MABL.  For example, during the SALTRACE campaign in
Barbados, lidar-derived LDR measurements within the lower MABL were $0.15 \pm 0.02$,
suggesting approximately equal parts spherical and non-spherical particles, despite in-situ
observations indicating surface dust mass concentrations as high as 40 $\mu g/m^3$ (Groß et al., 2016;
Weinzierl et al., 2017). Groß et al. (2016) also reported that dust mass concentrations exceeding
40 $\mu g/m^3$ could be underestimated by up to 50% by lidar-derived depolarization measurements,
in part due to the dominant influence of sea spray in the MABL that introduces large
concentrations of hydrated, spherical particles that reduce the overall depolarization signal.
Tsamalis et al. (2013) emphasized that the polluted dust aerosol type is often misclassified or
detected less often in spaceborne CALIOP observations due to low depolarization signals
resulting from dust mixing with other aerosol types such as biomass burning, marine or
anthropogenic aerosols (Yang et al., 2022; Kong et al., 2022). The relationship between dust
mass and depolarization has important implications for how the depolarization ratio is used to
infer surface-level dust concentrations in air quality forecasts and climate models. Since satellite
retrievals and column-integrated techniques lack vertical resolution, they may fail to capture
such near-surface morphological changes in dust (Li et al., 2020). If depolarization-based
methods underestimate dust presence near the surface under marine conditions, it could
introduce systematic errors in dust-related radiative forcing and deposition estimates.  A similar
concern exists for multi-angle imagers and polarimetric retrievals that depend on assumptions of
particle asymmetry to detect and quantify dust.

During August 2023, the Office of Naval Research (ONR) initiated the Moisture and

Aerosol Gradients/Physics of Inversion Evolution (MAGPIE) field campaign at the University of
Miami's Barbados Atmospheric Chemistry Observatory (BACO) at Ragged Point, Barbados to
map the inhomogeneity of the MABL. Central to MAGPIE are studies to identify information
lost when one conceptualizes the MABL as a series of uniform layers (e.g., surface layer, mixed
layer, entrainment or detrainment zones, etc.). While MAGPIE's core objectives focus on
atmospheric flows and fluxes with an emphasis on active remote sensing, aerosol particles and
their optical closure were implicitly a core mission element because light scattering by these
particles can be used to track atmospheric motion. MAGPIE collaborated across U.S. federal
agencies, academic institutions, and the Caribbean Institute for Meteorology and Hydrology
(CIMH) and included observations from ground-based aerosol particle samplers and instruments
at BACO along with local flights from the Naval Postgraduate School (NPS) CIRPAS Twin Otter
(CTO) aircraft. Central to the mission was the University of Wisconsin Space Sciences and
Engineering Center's (SSEC) High Spectral Resolution Lidar (HSRL; Eloranta et al., 2008).
Here, single particle and bulk analyses are used to evaluate how measured dust and sea salt mass
concentrations relate to HSRL-derived particulate LDR.  In Section 2, we provide a brief
overview of measurements, and in Section 3.1 a timeseries analysis of particle and lidar data,
demonstrating nonlinearity between dust and sea salt mass ratios to lidar LDR. In Sections 3.2
and 3.3, we provide vertically resolved single particle data from the CTO aircraft and ground-
based samples, respectively, to help explain the anomalies. In Section 4, we provide a discussion
and study conclusions.
**2.   Methods and materials**
**2.1. Sampling Site and Campaign Overview**
Ground-based aerosol particle and lidar measurements were conducted at the BACO site
on Ragged Point (13°6′N, 59°37′W) for August 2023. Situated at the easternmost point of the
Caribbean, BACO offers an optimal location for intercepting long-range transported Saharan
dust with minimal interference from local, anthropogenic emissions due to the prevalent Easterly
trade winds (Prospero et al., 2021; Gaston et al., 2024; Zuidema et al., 2019). Continuous aerosol
particle measurements have been conducted there for over 50 years, providing a unique long-
term observational record. The site is equipped with a tower that is 17 m high and is placed atop
a 30 m high bluff giving an altitude of ~50 m above sea level. While the measurements are not
taken directly at ground level, they are representative of the near surface MABL and are
routinely referred to as surface observations in prior Barbados studies (e.g., Zuidema et al.,

2019).

MAGPIE leveraged multi-platform measurements including aerosol particles collected at
the top of the BACO sampling tower and aboard the CTO aircraft to investigate vertical
gradients in aerosol particle chemical and morphological properties.  For the 2023 campaign, the
focus is centered around the largest dust events of the year observed between August 11-18,
2023. A total of five research flights were conducted during this period, with two samples
collected per flight, resulting in ten samples covering a range of altitudes from 30 m to 3 km
AMSL.
**2.2. Dust Mass Concentration Measurement**
Aerosol particles were collected on top of the BACO tower using high-volume samplers
with Total Suspended Particulate (TSP) inlets and fitted with cellulose filters (Whatman-41, 20
μm pore size) with a particle size cutoff at 80-100 μm in diameter due to the geometry of the
rainhat as described in Royer et al. (2023). Procedural filter blanks were collected every five
days and processed alongside the daily filter samples. A quarter of each filter was sequentially
extracted three times using a total of 20 mL of Milli-Q water to remove soluble components.
Following extraction, the filters were combusted at 500 °C overnight in a muffle furnace. The
residual ash mass was weighed and corrected for background contributions by subtracting the
ash mass obtained from the procedural blank. The net ash mass was multiplied by a correction
factor of 1.3 to account for the loss of any soluble or volatile components during the extraction
and combustion process (Prospero, 1999; Zuidema et al., 2019). While some soluble components
such as halite may be lost during the extraction process, the applied correction factor of 1.3 is
intended to conservatively account for these potential losses, supporting more robust dust mass
estimates. Moreover, halite is not a major constituent of Saharan dust, as previous studies report
its contribution rarely exceeds 3% by weight (Scheuvens et al., 2013), making any bias from its
loss during the extraction process unlikely to be significant.
**2.3. Sea Salt Concentration Measurement**
The filtrate collected after dust extraction on the daily filter samples and procedural
blanks was then analyzed using ion chromatography (IC; Dionex Integrion HPIC System;
Thermo Scientific). The samples were analyzed in triplicate for cations and anions and corrected
for procedural blanks. Details of our IC analysis procedure can be found in Royer et al. (2025).
Sodium ($Na^+$) is commonly used as a conservative tracer for sea spray particles, therefore, the
$Na^+$ concentrations measured by IC analysis were converted to equivalent sea salt concentrations
by applying a multiplication factor of 3.252 (Eqn. 1) (Gaston et al, 2024; Prospero, 1979).
$$\text{Sea salt concentration} = [Na^+]* 3.252 \qquad \text{Eqn. 1}$$
**2.4. In-situ Ground-based Aerosol Optical Measurement**
BACO is part of NASA's AErosol RObotic NETwork (AERONET). We used AERONET
level 2 aerosol optical depth (AOD at 500 nm) and fine mode AOD (at 500 nm) from the
AERONET spectral deconvolution retrieval (O'Neill et al., 2003) to identify the times of dust
intrusion during the sampling campaign (Giles et al., 2019; Holben et al., 1998).
**2.5. Single-Particle Analysis and Mixing State**
Aerosol particle mixing state describes how chemical species are distributed across the
particle population (Winkler, 1973; Riemer et al., 2019). Single-particle analysis offers a
powerful approach for analyzing this complexity, providing direct insight into the internal
composition and variability of individual particles (Reid et al., 2003a; Ault et al., 2014, 2012;
Royer et al., 2023; Casuccio et al., 1983; Kim et al., 1987; Andreae et al., 1986; Zhang et al.,
2003; Levin et al., 2005; Kandler et al., 2018). We used computer-controlled scanning electron
microscopy (SEM, Quanta from Thermo Fisher Scientific, equipped with a FEI Quanta digital
field emission gun at 20 kV and 480 pA electron current) coupled with energy-dispersive X-ray
spectroscopy (EDX, Oxford UltimMax100) (CCSEM/EDX) at the Environmental Molecular
Sciences Laboratory (EMSL) located at the Pacific Northwest National Laboratory (PNNL) to
characterize single particles. EDX spectra are collected for semi-quantitative analysis of the
particle elemental composition, and our analysis focused on 16 elements commonly found in
atmospheric aerosol particles: carbon (C), nitrogen (N), oxygen (O), sodium (Na), magnesium
(Mg), aluminum (Al), silicon (Si), phosphorous (P), sulfur (S), chlorine (Cl), potassium (K),
calcium (Ca), vanadium (V), manganese (Mn), iron (Fe), and nickel (Ni). This analysis was
conducted for particles collected on the BACO tower and aboard the CTO aircraft.
2.5.1.  **Ground-based Particulate Samples for Single Particle Analysis**
Ambient aerosol particles were sampled on top of BACO's 17 m tower using a three-stage
cascade impactor (Microanalysis Particle Sampler, MPS-3; California Measurements, Inc.), that
separates particles into aerodynamic diameter ranges of 2.5-5.0 µm (stage 1), 0.7-2.5 µm (stage
2), and 0.05-0.7 µm (stage 3). Samples were collected for 30 minutes at 2 L/min each day.
Particles were deposited onto carbon-coated copper grids (Ted Pella, Inc.) and analyzed using
CCSEM/EDX. No conductive coating (e.g., gold or carbon) was applied to the samples collected
on the ground as the conductivity of the copper grid bars minimized possible impacts from
charging effects. However, Cu signals from CCSEM/EDX were excluded due to interference
from the substrate. In contrast, C films are thin and highly transparent to electrons. Although C
signals are present in all spectra due to the support film, the C layer is fine-grained and
minimally interferes with particle morphology. Moreover, C together with O, serves as a useful
qualitative indicator for identifying organic particles, defined by a combined C + O contribution
exceeding 95 %. In this study, N was not used for quantification, nor did we label it in the EDX
spectra of particles. Elemental signals were considered valid for further analysis only when
exceeding a 2% threshold composition detected by EDX spectra. Over 1,000 individual particles
were analyzed per sample. Post-processing of CCSEM/EDX data was conducted using a k-
means clustering algorithm (Ault et al., 2012; Shen et al., 2016; Royer et al., 2023) to group
particles by similarity in composition and morphology. Clusters were classified into particle
types primarily based on semiquantitative elemental composition obtained from EDX analysis,
supported by particle size, morphology, and comparison with prior studies. Mineral dust particles
were identified by the presence of aluminosilicate elements (Si, Al, and Fe) characteristic of
crustal minerals (Hand et al., 2010; Krueger et al., 2004; Levin et al., 2005; Krejci et al., 2005;
Denjean et al., 2015). Fe was detected in ~80 % of mineral dust particles at relative area
abundances of 10-15 %. Sea spray particles were characterized by strong Na and Cl peaks,
indicative of halite (NaCl) and confirming their marine origin (Bondy et al., 2018). Aged sea
spray particles were identified by Cl depletion accompanied by enrichment in S, consistent with
heterogeneous reactions that replace Cl with sulfate or nitrate (Ault et al., 2014; Royer et al.,
2023, 2025). Mineral dust particles were observed to be both internally mixed with sea spray and
externally mixed (Royer et al., 2023, 2025; Kandler et al., 2018; Harrison et al., 2022; Aryasree
et al., 2024). These internally mixed dust and sea spray particles exhibited heterogeneous
compositions containing both dust-derived (Si, Al, Fe, Mg) and marine-derived (Na, Cl, Mg)
components, with Mg potentially originating from both sources. Organic particles were
dominated by C and O (>95 %), with minor inorganic elements, and typically appeared as
spherical or gel-like structures. Some displayed Mg-rich shells with sea salt cores, consistent
with primary marine organics formed via bubble-bursting at the ocean surface (Ault et al., 2013;
Gaston et al., 2011; Chin et al., 1998). Sulfate-rich particles exhibited strong sulfur peaks with
accompanying C and O signals, indicative of marine secondary aerosols (e.g., ammonium sulfate
or bisulfate) and frequently contained an organic fraction (O'Dowd and de Leeuw, 2007; Royer
et al., 2023).
2.5.2. **Airborne Particulate Samples for Single Particle Analysis**
Aerosol samples were also collected onboard the CTO using an isokinetic inlet and deposited
onto isopore membrane filters (47 mm filter, 0.8 µm pore size). An overview of the airborne
sampling technique can be found in the Supporting Information (SI Text S1). The CTO's primary
inlet has an intrinsic 50 % cutpoint of ~3.5 µm in aerodynamic diameter.  Due to limitations
associated with Teflon filter material, automated computer-controlled SEM was not feasible, and
these airborne samples were analyzed manually using SEM/EDX. To prevent particle charging
during imaging, filters were sputtered with a gold-coating of 10 nm thickness prior to analysis. A
total of 40, 21, and 52 particles from 250 nm to 25 µm diameter were manually analyzed from
samples collected within the SAL, above, and below cloud base heights (CBH), respectively,
providing a primarily qualitative assessment. The CBH was identified for each flight as the first
maximum in profile relative humidity, typically near saturation. Details of airborne sample
collection date and times, durations, altitudes, and corresponding CBH are provided in Table S1.
Particles were selected randomly across the filter area without targeting specific particle types or
sizes to reduce selection bias. All filter handling was performed in a laminar flow hood, and
filters were stored individually in sealed Teflon-taped Petri dishes to avoid any contamination.
The number of particles analyzed is reported in Table S2 of the SI. To quantify statistical
uncertainty, we calculated 95% confidence intervals for the number fraction of each particle class
assuming binomial sampling. The major particle types show varying levels of statistical
precision. For example, mineral dust is clearly dominant in the SAL ($90 \pm 9$ %) and statistically
distinct from mixed dust and sea spray particles, whereas above cloud top and below cloud base,
mineral dust and internally mixed dust and sea spray fractions have overlapping confidence
intervals, indicating comparable abundance within uncertainty. Thus, while the data robustly
supports dust dominance in the SAL, compositional differences among dust and dust mixed with
sea spray particle types in the above cloud top and below cloud base should be interpreted
qualitatively. Whenever possible, ground-based measurements were coordinated to coincide with
periods when the CTO aircraft intercepted the BACO location or its vicinity. Single particle
analysis from aircraft sampling, presented in Section 3.2, serves as a comparative reference to
the more comprehensive in-situ ground-based dataset, which includes ~24,000 analyzed
particles. The sulfate and organic particle types were absent in the airborne samples. This is
likely due, in part, to the use of isopore filters with a relatively large pore size (0.8 μm), that may
have limited the collection efficiency of finer sulfate and organic rich particles.
**2.6. High Spectral Resolution Lidar (HSRL)**
The SSEC HSRL was deployed during the summer 2023 MAGPIE campaign to
characterize the vertical distribution of aerosol particle scattering properties over Ragged Point.
The HSRL system used in this study can provide range-resolved profiles of particulate
backscatter and depolarization at high spatial and temporal resolution. Details on the SSEC
HSRL can be found elsewhere (Razenkov, 2010; Eloranta et al., 2008; Reid et al., 2025). Briefly,
the SSEC HSRL operates at a wavelength of 532 nm and separates molecular and particulate
backscatter signals using a narrowband iodine absorption filter. This configuration enables
accurate, independent retrievals of particulate backscatter ($m^{-1}$ $sr^{-1}$) within close proximity to the
ocean surface, as well as calibrated extinction ($m^{-1}$) and extinction-to-backscatter ratio (i.e., the
lidar ratio) measurements. The HSRL also contains an elastic backscatter channel of 1064 nm.
Long term Raman lidar measurements from the Max Planck Institute (MPI) in Barbados
(Weinzierl et al., 2017; Groß et al., 2015; Stevens et al., 2016) provides historical context for
aerosol backscatter and depolarization over the island and show structures consistent with the
HSRL observations presented here.
For MAGPIE, the SSEC-HSRL was configured to operate in periods of vertical stare,
horizontal stare, and vertical scanning from -0.05º to 18º. For the purposes of this paper, we only
utilize vertical data. Extraction of light extinction and the lidar ratio within the MABL are
performed using the HSRL in one of its side or vertically scanning modes. While a manuscript is
under preparation (Fu et al., 2025, in prep.), for the purpose of this paper we can report from its
authors that lidar ratios in the MABL's mixed layer ranged from 15 to 25 sr, and in the SAL was
on the order of 35-40 sr. Lidar ratios of 15-20 sr are consistent with ambient sea salt (RH= 70-
85% near the surface) and 40 sr above the MABL for "dry" dust in the less humid SAL (RH= 30-

50%).

**3.   Results and discussion**
**3.1. Temporal variability in surface-level aerosol particle chemistry, AOD and lidar**
**depolarization ratios (LDR) during a major dust event**
Figure 1 presents a time series of key aerosol properties observed during the August 2023
MAGPIE intensive operations period, including surface-level dust and sea salt mass
concentrations, aerosol optical depth (AOD), and HSRL-derived particulate linear depolarization
ratio (LDR) and particulate backscatter. Over the month, median dust and sea salt concentrations
were $6 \pm 32$ and $17 \pm 9$ µg/m$^3$, respectively; the median columnar AOD was $0.15 \pm 0.19$; and the
median LDR at 135 m AMSL was $0.02 \pm 0.03$. Notably, a distinct deviation from these baseline
values was observed during a period of Saharan dust intrusion occurring between August 11 and
18, 2023. The dust event led to pronounced changes in the chemical composition and physical
properties of aerosol particles observed in Barbados, yet the LDR showed little increase. During
this period, the dust mass concentration peaked at 120 µg/m$^3$ on August 15, comparable to the
concentration measured during the major "Godzilla" dust event of 2020 (Elliott et al., 2024;
Mayol-Bracero et al., 2025), while inferred sea salt concentrations based on sodium were 27
µg/m$^3$, representing an upper-limit estimate given the possible contribution of Na from mineral
dust. The average dust-to-sea salt mass ratio was ~3.4 on dusty days (peaking at 4.8), compared
to ~0.40 on non-dusty days, indicating a clear dominance of dust in the lower MABL during the
dust intrusion event. Total column AOD (550 nm) closely tracked the trend in surface dust mass
concentration and peaked at ~0.75 on August 15, whereas fine mode AOD remained
substantially lower ($0.12 \pm 0.01$; Fig. 1b) indicating that the total AOD was predominantly
influenced by coarse-mode particles during the dust period. Notably, this event produced one of
the highest AOD recorded in Barbados during the month of August over the past decade (Fig.
S1).

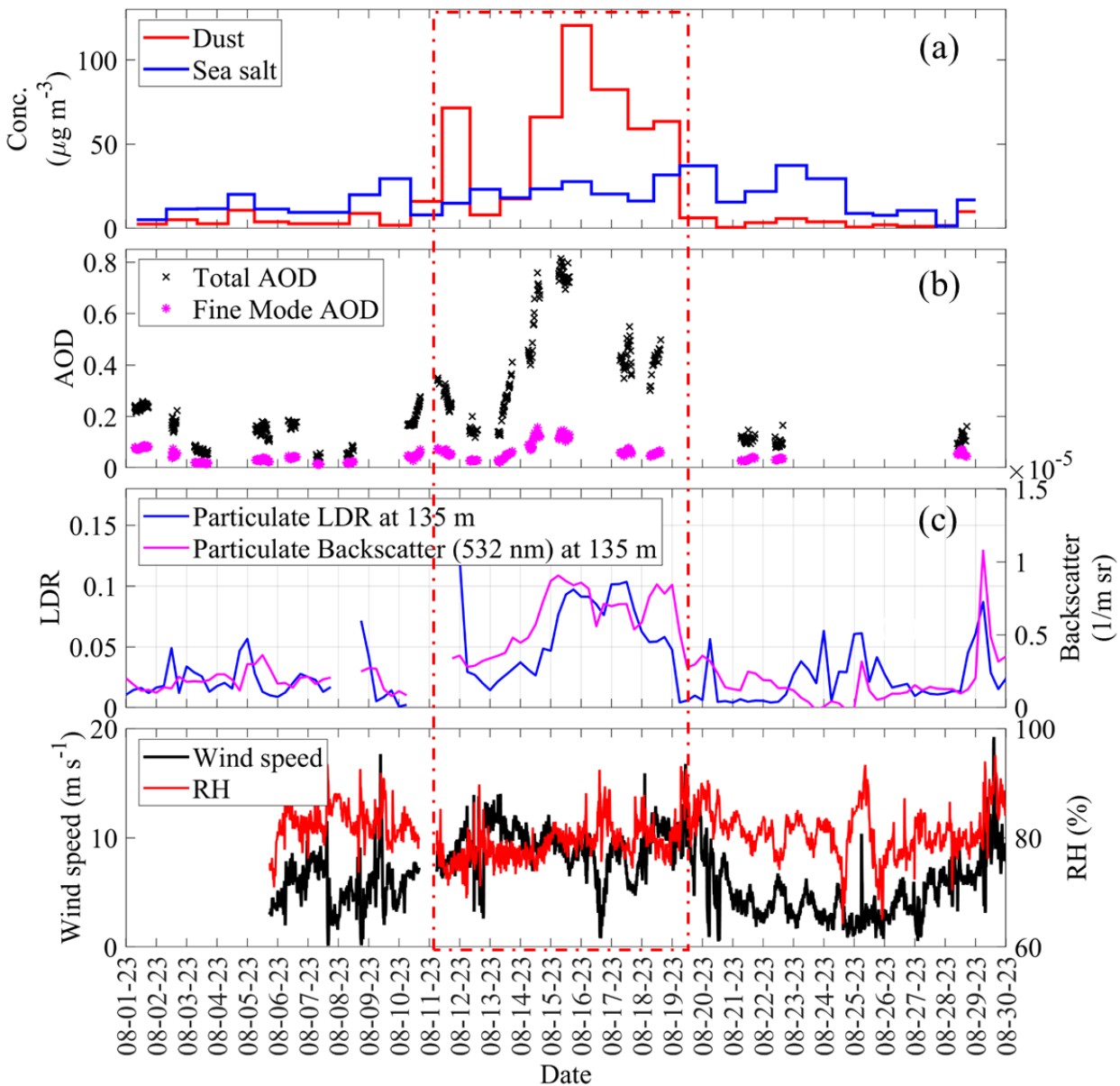


Figure 1. Time series plots for (a) dust and sea salt mass concentrations measured from the top of

the BACO tower, (b) AERONET total column and fine mode fraction AODs (at 500 nm), (c)

HSRL- particulate linear depolarization ratio (LDR) and particulate backscatter at 532 nm,

averaged over six hours, and (d) meteorological measurements (RH and wind speed) during the

MAGPIE 2023 campaign. The red dashed box represents the major dust intrusion periods

observed during the campaign.

Figure 1c presents the time series of the particulate backscatter and LDR at 135 m
AMSL, representing conditions near the surface within the lower MABL for comparison with
other ground-based measurements. Although an increase in LDR was observed in the lower
MABL during the period of pronounced dust loading, the enhancement was surprisingly small,
with values of 0.10 or less (Fig. 1c).  The finding can be partially explained through scattering
physics (e.g., the lidar equation) governing the lidar signals (Hayman and Spuler, 2017). For
MAGPIE, the HSRL lidar ratio (LR), the ratio of aerosol extinction ($m^{-1}$) to backscatter ($m^{-1}$ sr$^{-}$
$^{1}$), was approximately 40 sr for dust and 20 sr for marine aerosols. Because LR is inversely
related to the particulate 180° backscatter phase function, a lower LR indicates that marine
aerosol particles scatter back approximately twice the amount of energy compared to dust if the
marine and dust extinctions are the same. This difference in backscatter directly affects the
measured LDR. In a mixed aerosol layer with comparable extinction from dust and marine
particles, the backscattered signal, on which the LDR is based, is weighed more strongly toward
the marine aerosol contribution (that has a lower LDR).
Given that dust concentrations were approximately four times greater than those of sea
salt during the peak of the event, we applied a multiple regression approach to estimate the LDR,
using Eqn. 2, that incorporated the measured lidar ratio and dust and sea salt concentrations.
$$LDRexpected = \frac{v_{\perp}^{(d)}}{v_{\parallel}^{(d)} + v_{\parallel}^{(m)}} + \frac{v_{\perp}^{(m)}}{v_{\parallel}^{(d)} + v_{\parallel}^{(m)}} \qquad Eqn.\,2$$

where, $v_{\parallel}^{(d)}$ and $v_{\parallel}^{(m)}$ represent the parallel components, and $v_{\perp}^{(d)}$ and $v_{\perp}^{(m)}$ represent the
perpendicular components of the particulate backscatter from dust ("d") and marine aerosol
("m") particles, respectively.
This analysis yielded an estimated LDR of $0.17 \pm 0.03$ during the dust peak, ~2 times higher
than the values observed in Fig 1c in the lower MABL. Details about this calculation and
approximations used to derive this estimate are in SI Text S3. Figure 2 shows the relationship
between the dust-to-sea salt mass concentration ratio versus the measured HSRL-derived LDR
and estimated LDR from the multiple regression approach. We note several caveats to our
calculation of the estimated LDR. First, the uncertainty associated with our estimated LDR
prediction may be larger than the standard deviation reported, as we did not explicitly account
for the full-size distribution of sea salt and dust aerosols. In particular, large particles beyond the
upper cut point (>80 -100 μm) of our bulk dust sampler were not captured. While previous
studies have shown that some particles of this size can survive trans-Atlantic transport (e.g.,
Betzer et al., 1988; Reid et al., 2003a; Barkley et al., 2021), their number concentrations are
expected to be substantially lower than those of the particle sizes efficiently collected by the
filter sampling used in this study. These coarse particles, which are more efficient at depolarizing
incident light due to their irregular shape and size, could contribute significantly to the lidar
signal. Their absence from the analysis may lead to an underestimation of the true depolarization
potential, especially during intense dust events. Nevertheless, we recognize that other factors
may also influence the observed reduction in depolarization. Vertical heterogeneity within the
MABL, including overlapping layers of marine and dust aerosols, could further convolute the
dust depolarization signal. In addition, inherent limitations in HSRL retrievals, such as signal
averaging in optically thin layers or reduced sensitivity near the ocean surface may contribute to
the apparent underestimation of LDR.

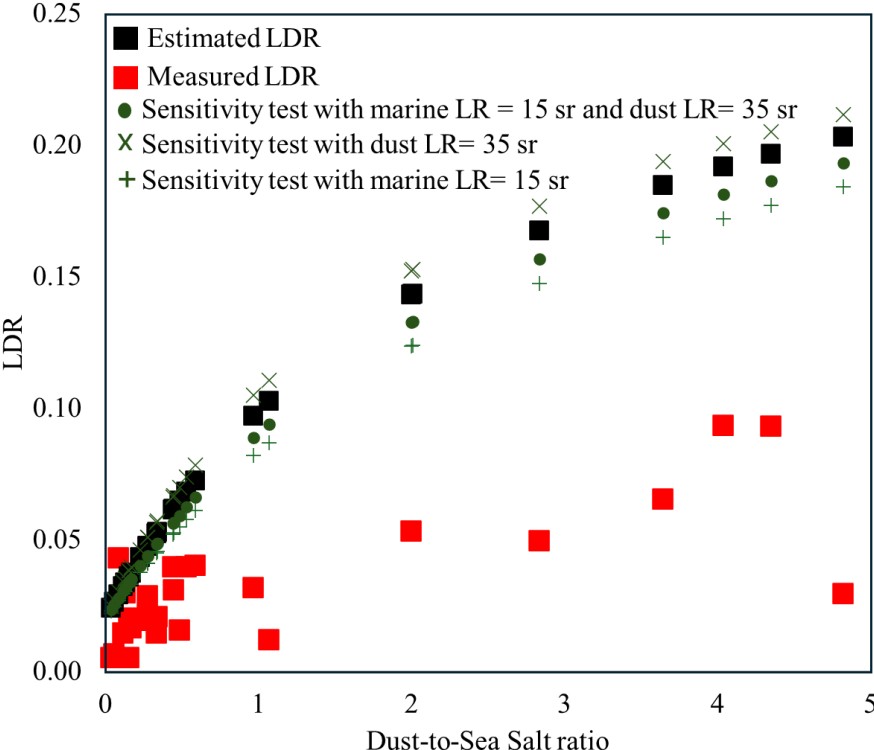


Figure 2. Relationship between the dust-to-sea salt concentration ratio and HSRL-derived

particulate LDR at 135 m above ground level during the MAGPIE campaign. Red squares

indicate measured LDR values for the full campaign, while black squares represent LDR values

estimated from mass concentrations and lidar ratio weighting during the peak dust event. The

calculated LDR was approximately a factor of two higher than what was observed during the

peak dust event. A sensitivity test was conducted using more conservative lidar ratio values for

dust and marine aerosols (shown as green plus, cross and circle symbols), and in all such cases

the estimated LDR values remained consistently higher than the measured values.

Extending our findings in Figure 1c vertically, Figure 3a and b shows the time series of

particulate backscatter and LDR measurements from August 12-18, 2023, at altitude up to 6 km

AMSL. At ~2-6 km AMSL above Ragged Point, measurements of increased particulate

backscatter (shown in Fig 3a) are primarily attributable to increased dust loading within the SAL,

as indicated by the concurrent elevated LDR of 0.30 (shown in Fig 3b). This altitude range is
consistent with previous studies that have reported the SAL to typically extend from
approximately 1.5 to 5.5 km AMSL (Carlson and Prospero, 1972; Groß et al., 2015; Karyampudi
and Carlson, 1988; Reid et al., 2003; Weinzierl et al., 2017). The particulate backscatter
measurement shown in Fig. 3a highlights high aerosol loading near the surface, consistent with
the large concentration of marine particles in the lower MABL. Notably, periods of enhanced
backscatter between August 14-16 extending downward from the SAL into the MABL suggest
episodes of dust downmixing toward the surface, which are also supported by a concurrent
increase in surface dust mass concentrations (Fig. 1).

Figure 3c shows the representative vertical distribution of RH during the dusty period of

the study, revealing a distinctly moist MABL characterized by RH values exceeding 80%. Such
elevated humidity levels are conducive to the hygroscopic growth of aerosol particles, which can
increase both particle size and sphericity (Titos et al., 2016). These changes in particle properties
caused by hygroscopic growth can further enhance particle backscatter while decreasing the
LDR which is visible in the particulate backscatter (Fig. 3a) and LDR (Fig. 3b) measurements
below cloud base (~700 m). Thus, under humid MABL conditions, both the LR contrast between
dust and marine aerosols and hygroscopicity-driven growth can act together to suppress the
observed LDR. However, a key consideration is aerosol mixing state as previous observations
have shown limited hygroscopic growth of African dust particles, even at high RH, but
substantial growth of dust particles that are internally mixed with other aerosol components
including sea spray (Denjean et al., 2015).

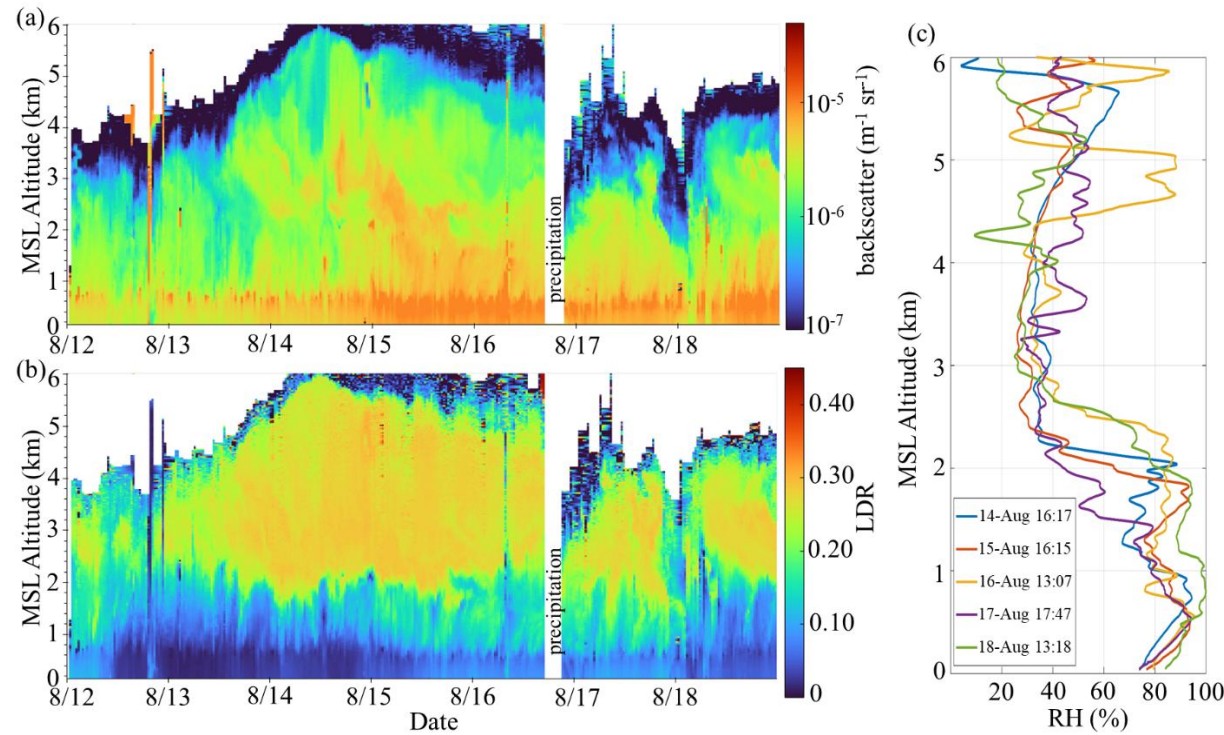


Figure 3. HSRL -measurements for (a) particulate backscatter (m$^{-1}$ sr$^{-1}$) and (b) particulate linear
depolarization ratio (LDR) within 6 km AMSL for August 12 -18, 2023. (c) Vertical profiles of
relative humidity (RH, %) up to 6 km AMSL from radiosonde launches at Ragged Point on
representative days between August 14 and 18, 2023. In panels (a) and (b), periods with
particulate backscatter <10$^{-7}$ (m$^{-1}$ sr$^{-1}$) are masked out. The uncertainty associated with the
particulate LDR measurements shown in panel (b) is provided in Fig. S2.
**3.2. Vertical Gradients in the LDR and aerosol mixing state**

A vertical gradient in aerosol particle mixing state was observed during the Saharan dust

intrusion, wherein dust is internally mixed with sea spray at the surface and externally mixed
aloft. Single-particle chemical composition and morphology analysis revealed a diverse set of
particle types with distinct chemistries and morphologies, including mineral dust, sea spray, aged
sea spray, internally mixed mineral dust and sea spray, sulfates, and organics (Royer et al., 2023;
Ault et al., 2012, 2014). The Methods section describes the particle classification approach and
the particle types identified in this study. Detailed chemical composition of the particle types is
presented in SI Text S2, representative elemental digital color stack plots used for particle
classification are shown in Fig. S3, and representative SEM images and corresponding EDX
spectra for each particle class are shown in Fig. 4a.

Our single particle results from ground-based samples share several similarities with, but

also important differences from, previous studies of Saharan dust transported to the Caribbean.
Consistent with Harrison et al., 2022; Krejci et al., 2005; Denjean et al., 2015 and Reid et
al.,2003a for the Caribbean, the vast majority of dust particles observed at Barbados during
MAGPIE were aluminosilicates, confirming the dominance of this mineralogical class in trans-
Atlantic Saharan dust. A prominent feature of the MAGPIE observations was the frequent
presence of internally mixed dust and sea spray particles, a phenomenon also documented in
earlier Caribbean studies (e.g., Reid et al., 2003a; Aryasree et al., 2024; Royer et al., 2025).
Kandler et al. (2018) suggested that such mixing likely occurs locally through turbulent
interactions between dust and marine aerosol in the MABL. Our observations are consistent with
this mechanism and further suggest that cloud processing may enhance this internal mixing.
Similar internally mixed dust and sea spray particles have been reported in other coastal regions,
particularly during Asian dust outbreaks (Zhang and Iwasaka, 2004; Zhang et al., 2006; Zhang
and Iwasaka, 2001; Zhang et al., 2003), indicating that this mixing process is not unique to the
Caribbean but may be characteristic of dust outflows across humid marine environments.

Figures 4c and S4 present the average size-resolved chemical composition of ground-

level aerosol samples collected during the dust event. A clear compositional shift is observed
between submicron and super-micron particles. In the submicron range (particle diameter <
1 µm), organic and sulfate aerosol particles were dominant, with median diameters of 0.45 µm
and 0.36 µm, respectively. In contrast, the super-micron size range was dominated by sea spray,
mineral dust, and internally mixed dust and sea spray particles. Externally mixed mineral dust
collected through our impactor had a number median diameter of ~1.2 µm, while internally
mixed dust and sea spray particles exhibited larger median diameters of ~2.0 µm, likely resulting
from coagulation and condensation processes occurring during dust descent into the MABL
(Kandler et al., 2018). Further, these particles likely become even larger under the high relative
humidity (>80 %) conditions of the MABL consistent with hygroscopic growth (Zieger et al.,
2017). This morphological evolution in internally mixed dust and sea salt particles would
explain, in part, the suppressed LDR during the major dust intrusion event (Bi et al., 2022).

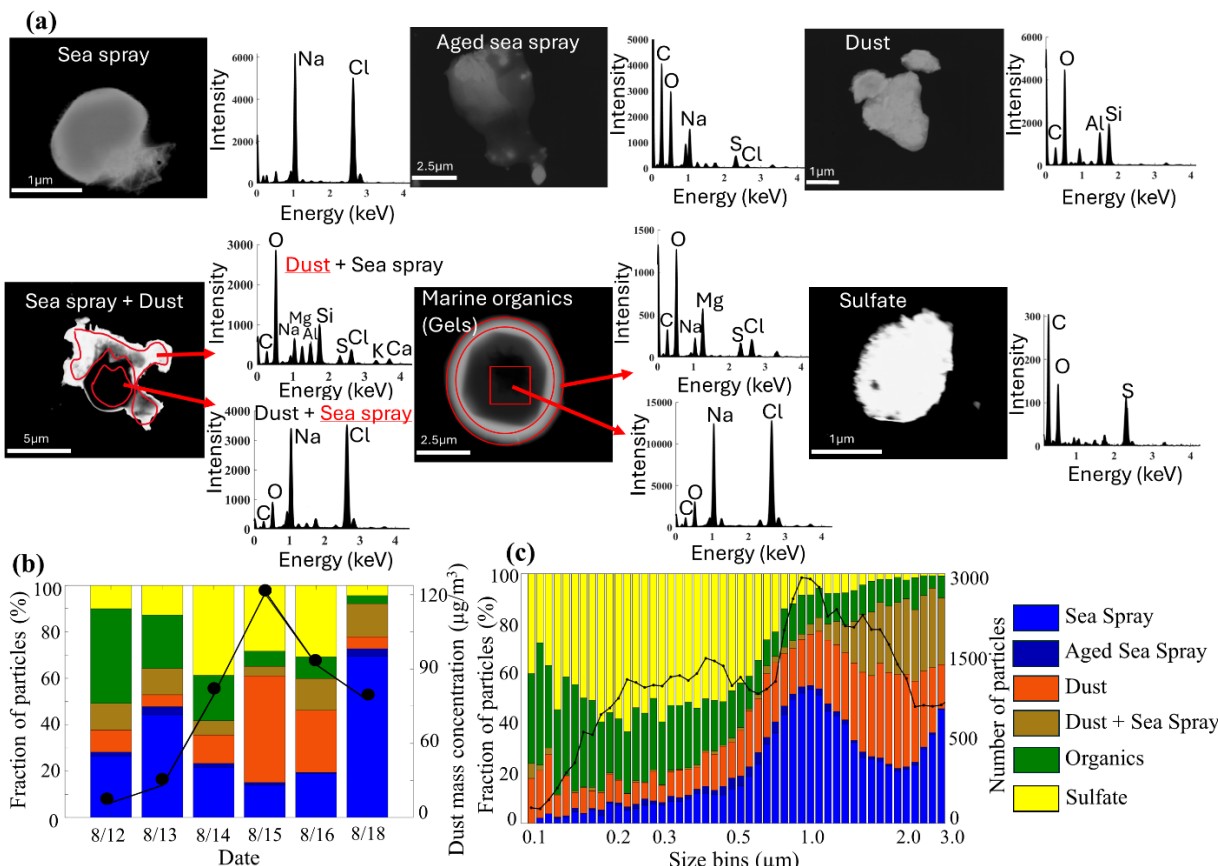


Figure 4. (a) Representative aerosol particle types observed in surface samples by SEM images
(left) and EDX spectra (right) in samples collected during the MAGPIE campaign. (b) Temporal
variations in the number fraction of different particle types during the dust event. (c) Number
fractions of different particle types plotted as a function of the particle projected area diameter.
The black colored line graph in panels (b) & (c) represents dust mass concentration and number
of particles, respectively. These plots are generated from the single particle CCSEM/EDX
analysis of the in-situ samples collected at the top of the 17 m tower at BACO.

To extend this analysis vertically and examine how particle composition varies with

altitude, Fig. 5 presents the vertical profile of the number fractions of aerosol particle types,
averaged over the samples taken during dusty days, and HSRL data from August 15 (15:00
UTC), the day when Barbados experienced the highest ground-level dust concentration (~120
µg/m$^3$). The SAL was predominantly composed of mineral dust particles (90% of the analyzed
particles) transported from Northern Africa, and the LDR observed within the SAL (0.30) is
attributable to the large fraction of mineral dust present in this layer. Additionally, a transition
layer between the SAL and the MABL (labeled as "Mixture" in Fig. 5b) is shown where both sea
salt and mineral dust are concurrently present. In the SAL, a fraction of the dust is internally
mixed with sea spray particles (10% of the analyzed particles). Below the SAL, between 0.7 km
and 1.8 km, LDR values were much smaller and ranged from 0.10 to 0.20, typical for aerosol
regimes within the humid MABL where mineral dust particles are mixed with sea spray particles
(Gasteiger et al., 2017; Tesche et al., 2011).  A comparison of particle composition across
altitudes reveals that samples collected above the cloud top contained a slightly higher number
fraction of mineral dust (57%) compared to internally mixed dust and sea spray particles (43%).
In contrast, below the cloud base, this ratio was reversed, with internally mixed dust and sea
spray particles making up 58% of the dust and externally mixed dust 42% of the dust particles
suggesting a dynamic, vertical exchange of particles within the MABL. The MABL circulation
pattern through clouds is well documented by lidar observations (e.g., from early studies (Kunkel
et al., 1977) to more recent work (Reid et al., 2025). Such cloud processing mechanisms likely
enhance coagulation while turbulent updrafts promote collisions between sea spray and dust
particles (Matsuki et al., 2010). The presence of a substantial fraction of internally mixed dust
and sea spray particles above and below cloud base is expected, given that sea salt is a dominant
contributor to cloud droplets (Crosbie et al., 2022). The number fraction of mineral dust particles
increased substantially in the MABL during periods of intense dust intrusion, with a distinct peak
observed on August 15 (Fig. 4b). However, particle composition was more variable at the surface
compared to aloft, consistent with the proximity to the ocean increasing the presence of marine
aerosol particles including sea salts, organics, and sulfates (Fig. 5c). Further, at altitudes below
0.7 km, LDR values were consistently at or below 0.10, commonly taken as being indicative of
the dominance of sea spray particles with reduced dust influence (e.g., "Dusty Marine" in the
CALIPSO retrievals; Kim et al., 2018).

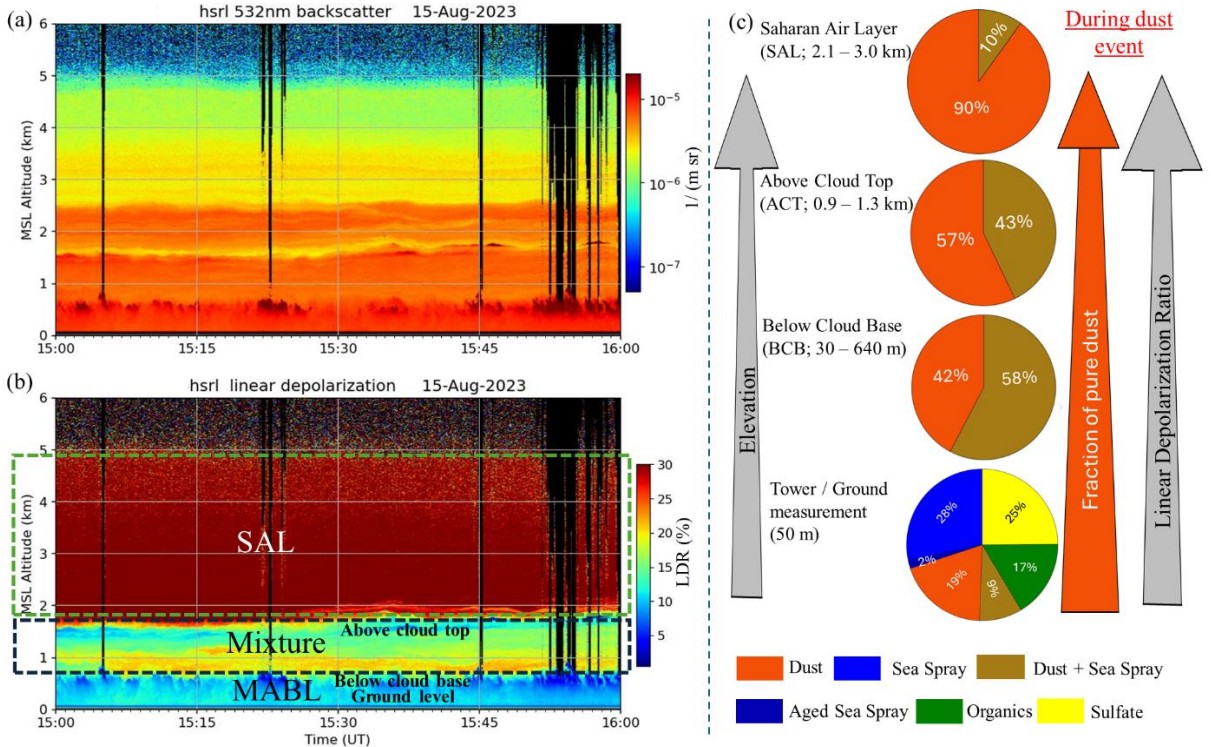

Figure 5. HSRL scan for (a) particulate backscatter at 532 nm and (b) particulate linear

depolarization ratio within 6 km AMSL for August 15, 2023 (15:00 hrs UTC). (c) Pie charts

showing the number concentration (as a percent) of particle types detected from single particle

analysis at different altitudes: SAL, above cloud top, below cloud top, and ground-based samples

collected atop the BACO tower during the dust event. The altitude range where samples

collected for single particle analysis were taken are indicated in parentheses next to each

corresponding pie chart. Pie charts show that with increased elevation, the fraction of externally

mixed dust increased and the linear depolarization ratio (LDR) from the HSRL measurement

increased during the dust event. The RH vertical profile from a radiosonde launched during this

HSRL observation period shown in panels (a) and (b) is shown as the orange line in Fig. 3c.

**3.3. Accounting for Dust Mixing State and Hygroscopic Growth in Predicting the LDR**

Prior work by Denjean et al. (2015) showed that externally mixed African dust did not

exhibit hygroscopic growth even at high RH (up to 95%), whereas appreciable water uptake
occurs primarily when dust is internally mixed with sea spray, a particle type that was
prominently observed in our single-particle analysis. Thus, we evaluated how the expected LDR
changes when RH-dependent optical weighting is explicitly accounted for by applying a
hygroscopic extinction enhancement factor to internally mixed dust and sea spray particles. The
detailed discussion of this hygroscopicity dependent calculation is provided in the SI Text S4,
and the resulting LDR predictions are shown in Fig. 6. The enhancement factor ($\chi$) represents the
marine aerosol extinction enhancement due to the increase in the marine particle cross-sectional
area with increasing RH (i.e., hydroscopic growth) (Hänel, 1972, 1976). When this enhancement
factor is included, the estimated LDR is further suppressed, consistent with our observations that
dust in the moist MABL becomes internally mixed and more spherical when hydrated. This
refined estimate improves closure between the measured and predicted depolarization ratios
suggesting that hygroscopic growth of internally mixed dust and sea spray particles play a central
role in reducing the lidar depolarization signal. Further, simulations of light scattering by
nonspherical particles and coated particle systems by Bi et al. (2022) showed that mineral dust
particles coated by a hydrated, low refractive index shell (e.g., water, sulfate, or sea salt) can
exhibit a strongly suppressed depolarization signal, often approaching values characteristic of
spherical particles. This occurs because at high RH the hygroscopic shell grows substantially and
dominates the optical response, effectively masking the non-sphericity of the underlying dust
core. This coated particle behavior could provide a physical basis for our observations in the
humid MABL, where internally mixed dust and sea spray particles observed at RH consistently
exceeding 80% produce low LDR values (<0.1) despite high dust mass concentrations and
highlights the need to investigate the role of particle composition and mixing state in modulating
depolarization signals. Overall, these observations suggest that the reduced LDR values in the
MABL are likely explained, in part, by internally mixed dust and hydrated sea spray particles in
the presence of high humidity, resulting in hydrated, more spherical and hence less depolarizing
particles.

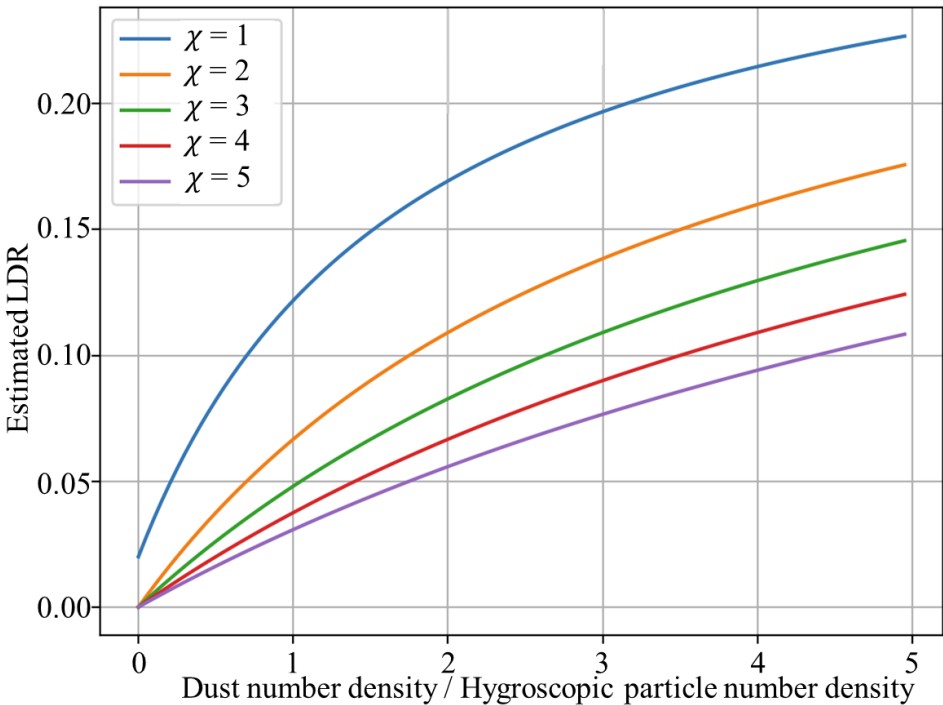


Figure 6. Relationship between estimated LDR and dust-to-hygroscopic particle number density
ratio as a function of marine aerosol extinction enhancement factor ($\chi$) due to hygroscopic
growth. The estimates are based on the observed HSRL-LDR for dry dust particles as 0.3 and LR
for dry dust particles as 35 sr and the observed average ratio of cross-sectional area of internally
mixed dust and sea spray particles to that of externally mixed dust particles as 2.7, derived from
CCSEM/EDX single-particle analysis of surface samples collected at BACO. Cross-sectional
areas were calculated using the respective median diameters measured for each particle type.

## 4. Conclusions and Atmospheric Implications

Single-particle analysis conducted during the MAGPIE campaign revealed that Saharan dust particles in the MABL are physically and chemically distinct from dust within the SAL aloft. Our results show that in the lower, humid MABL, dust becomes internally mixed with sea spray resulting in potentially enhanced hygroscopicity compared to externally mixed dust in agreement with prior studies investigating the hygroscopicity of transported African dust (Denjean et al., 2015). These changes, in part, suppress the dust's depolarization (being more spherical) signal and complicate its identification by lidar. Despite peak dust loading at BACO (AOD ~0.75; surface dust ~120 µg/m³), HSRL observations showed that LDR values in the lower MABL remained mostly below 0.10, a range typically associated with spherical marine aerosols, even though dust concentrations were ~4.8 times higher than sea salt. This discrepancy is further explained by differences in the scattering (lidar ratio) of dust and marine aerosols, where dust backscatters half the energy per extinction cross-section (lidar ratio) compared to marine aerosols which lowers the depolarization measurement. These combined effects of morphological transformation and different lidar ratios reduce the dust signature in depolarization-based retrievals, complicating its detection and quantification near the surface. The resulting underestimation of surface-level dust by lidar-based depolarization retrievals is of particular concern especially during high-dust events like the one observed during this study, where surface particulate matter (PM) exceeded WHO guidelines for $PM_{10}$ of 45 µg/m$^3$ (World Health Organization, 2021) by a factor of nearly three. Moreover, it may help explain similar discrepancies between lidar observations and in situ measurements in other regions where dust is modified through interactions with marine aerosols.

More broadly, these results highlight the importance of integrating vertically resolved
lidar data with in-situ single-particle analysis and surface aerosol mass concentrations to improve
the interpretation of lidar observations in dust-affected regions. Such integrated approaches are
essential because LDR is widely used in satellite retrieval algorithms and atmospheric models to
estimate dust volume and mass fractions, calculate dust-related radiative forcing, estimate dust
contribution to cloud condensation and ice nucleation profiles, estimate dust deposition to
receptor ecosystems, and predict surface air quality (Meloni et al., 2018; Haarig et al., 2017;
Müller et al., 2010, 2012; Yang et al., 2012; Marinou et al., 2017; Proestakis et al., 2018; Adebiyi
et al., 2023; Mahowald et al., 2005). Without such integrated observations, satellite retrievals and
forecasting systems may significantly underestimate dust impacts near the surface, where they
matter most for air quality and biogeochemical feedback.
While our results demonstrate that single wavelength depolarization can underestimate
near surface dust under humid, mixed aerosol conditions, we emphasize that more advanced
remote sensing approaches can mitigate these limitations. Multi-wavelength HSRL observations,
including backscatter at 532, and 1064 nm and corresponding color ratio and depolarization
metrics, provide additional degrees of freedom for discriminating dust from hydrated marine
aerosol particles. In fact, recent upgrades by the SSEC HSRL team have produced the first
calibrated 1064 nm HSRL system, that is aimed at being deployed in future studies. These multi-
spectral measurements would enable color ratio signatures characteristic of dust to be detected
even when LDR is low, thereby providing a remote sensing pathway to constrain surface dust
loading. Validating these multi-spectral retrievals requires independent constraints on aerosol
composition and morphology. The vertically resolved single particle measurements presented
here provide validation of how dust properties change as they mix with sea spray. Thus, rather

than diminishing the utility of lidar, our results highlight the importance of integrating advanced

multi-wavelength lidar products with targeted in-situ observations to improve the accuracy of

surface dust estimates in marine environments.

**Data Availability**

Dust and sea salt mass concentration data and number counts of particle types detected by

CCSEM/EDX is publicly available in the University of Miami data repository

(https://doi.org/10.17604/1427-0558).

The HSRL data can be accessed through the University of Wisconsin-Madison SSEC repository

at https://hsrl.ssec.wisc.edu/by_site/37/bscat/2025/04/.

The NASA AERONET data can be accessed through https://aeronet.gsfc.nasa.gov.

**Author Contribution**

Conceptualization of this work was done by SS, RJH, JSR, and CJG. JSR posed the initial

hypothesis and designed the data collection strategy. Collection of samples was conducted by SS,

WJM, ZB, IR, EE, JSR, EB, ADO, RCL, AA, DB, EAR, JRP, AB, RY, QW, TE, EL, MLP, and

CJG, while analysis was done by SS, HEE, NNL, ZC, SC, and RA. The development of method

used in this work was done by SS, REH, WJM, EE, JSR, and CJG. Instrumentation used to

conduct this work was provided by REH, SC, MLP, and CJG. Formal analysis of data was

performed by SS, WJM, and JSR. EE performed the optical calculations of expected LDR.

Validation of data products was performed by SS, RJH, WJM, JSR, AA, and CJG. Data

visualization was performed by SS. Supervision and project administration duties were done by

RJH, JSR, and CJG. SS wrote the original draft for publication, and all the co-authors reviewed

and edited this work.

**Competing Interests**
The contact author has declared that none of the authors has any competing interests.
**Acknowledgements**
We thank the family of HC Manning and the Herbert C Manning Trust for providing access to
their land at Ragged Point in Barbados. We thank Jeremy Bougoure at EMSL for his help with
the Au sputter coating of our filter samples. We thank Dr. Konrad Kandler and the other
reviewers for their constructive and insightful comments, which substantially improved the
clarity and rigor of this manuscript.
**Financial Support**
CJG and SS acknowledge the Office of Naval Research (ONR) grants N00014-23-1-2861 and
N000142512003 and NSF MRI grant 2215875.  A portion of this research was performed on
project awards (10.46936/lser.proj.2021.51900/60000361 and
https://doi.org/10.46936/ltds.proj.2023.61072/60012372) from the Environmental Molecular
Sciences Laboratory, a DOE Office of Science User Facility sponsored by the Biological and
Environmental Research program. REH, WJM, ZB, IR and EE were supported under ONR grant
N000142412736. JSR and EAR were supported under ONR grant O2507-017-017-112205. AB
was supported under ONR grant N0001423WX01787. QW, JRP and RY were supported under
ONR grant N0001424WX02429. APA acknowledges support from ONR grant N000142512003
and DOE grant DE-SC0025196.

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
