# Peer review of "Transported African Dust in the Lower Marine Atmospheric Boundary Layer is Internally"

_EGUsphere, 2025_

## Referee Comment (RC4)

**Comments on**

**Transported African Dust in the Lower Marine Atmospheric Boundary Layer is Internally Mixed with Sea Salt Contributing to Increased Hygroscopicity and a Lower Lidar Depolarization Ratio**

This study investigated the evolution characteristics of physical and chemical properties of Saharan dust during its interaction with sea salt aerosols during long-range transport, and carefully analyzes the vertical distribution characteristics of the aging process of dust aerosols in the marine atmospheric environment, which is of great importance for understanding the radiative properties of dust aging. The results also highlight the importance of integrating vertically resolved lidar data with in-situ single-particle analysis and surface aerosol mass concentrations to improve the interpretation of lidar observations in dust-affected regions. The manuscript's presentation of linguistic logic is clear and rigorous, and its overall writing quality is good. Nevertheless, there are some minor issues in the manuscript that require further revision and clarification. Only when the following issues have been revised or clarified is it recommended for publication:

- (1) The manuscript contains a large number of abbreviations, and it is recommended to include a list of abbreviations.
- (2) Section 2.2 and Figure 1a: How is the dust mass concentration derived?
- (3) Section 2.2: The method for calculating sea salt concentration is based on the assumption that all Na+ originates from sea salt aerosols. However, dust aerosols contain a certain amount of sodium salts. Although the proportion of Na+ in dust is small, when the dust concentration is high, the Na+ contribution from dust may be difficult to ignore. Therefore, it is recommended to recalculate the sea salt concentration after deducting the Na+ from dust aerosols.
- (4) Line 247: The part after "0.03" lacks a period.
- (5) It is recommended to add meteorological data on Figure 1, such as temperature, relative humidity (RH), wind speed/direction. In particular, RH can assist in understanding the role that meteorological conditions within dust plumes played in altering the physicochemical

- properties of aged dust (dust + sea salt) with high hygroscopicity.
- (6) The interaction between dust aerosols and sea salt over the ocean has long been studied. For instance, about 20 years ago, Zhang et al. investigated the interaction between dust aerosols originating from the Asian continent in East Asia and sea salt aerosols in the northwestern Pacific Ocean. It is recommended that the authors compare this study with previous research to highlight the innovations of this paper. (References: Zhang, D. Z., et al., Geophys. Res. Lett. 2001, 28 (18), 3613-3616; Zhang, D. Z., et al., Mixture state and size of Asian dust particles collected at southwestern Japan in spring 2000. J. Phys. Chem. A 2003, 108 (D24); Zhang, D. Z. and Iwasaka, Y., Size change of Asian dust particles caused by sea salt interaction: Measurements in southwestern Japan. Geophys. Res. Lett. 2004, 31 (15); Zhang, D. Z., et al., Coarse and accumulation mode particles associated with Asian dust in southwestern Japan. Atmos. Environ. 2006, 40 (7), 1205-1215.)
- (7) Figure 5b shows that when dust concentration is relatively high, the fraction of sulfate particles is also high. Why? Are these sulfate particles derived from anthropogenic emissions or natural sources? Is there a possibility that these sulfate particles originate from dust aerosols? Recent studies have shown that fresh dust aerosols also contain sulfate (Li, W. et al., A Review of Water-Soluble Ions in Natural Dust Particles Over East Asia: Abundance, Spatial Distributions, and Implications. ACS ES&T Air 2025, 2 (8), 1379-1393).

---

## Author Response (AR1)

Reviewer comments are presented in black text, and our responses in blue text. Added text to the manuscript are *italicized*. Each comment is labeled using the notation **R#.C#**, where **R#** denotes the *reviewer number* and **C#** denotes the *comment number*. The line numbers referenced in this document correspond to the clean copy version of the revised manuscript.

**Reviewer #1**

This paper focuses on field measurements during ONR's MAGPIE campaign in August 2023 at Ragged Point, Barbados. It focuses on how Saharan dust mixes with sea salt, yielding surprisingly higher hygroscopicity than what would normally be expected for dust. Of course this can be explained by the mixing with sea salt. This act of mixing appears to be localized in the bottom altitudes unlike higher up in the Saharan Air Layer. The enhanced hygroscopicity and mixing at lower altitudes coincides with reduced HSRL-derived depol ratios in contrast to higher altitudes where there is just dust. The topic is important and points to important nuances in the transport of dust and how we really must understand its mixing behavior with other aerosol types to know the chemical, physical, and optical properties of such air masses. I believe the paper will have interest among the readers of the journal.

The methods used are robust and include measurements of single particle properties, concentrations of sea spray and dust, and HSRL retrievals.

The presentation quality is good. In general, the paper was well done and publication is recommended. Some minor suggestions/questions are provided below.

We appreciate the reviewer's positive assessment and recommendation for publication. Detailed responses to the reviewer's comments are addressed below.

Specific Comments:

**R1.C1:** Data Availability section: a bit odd to say "will be publicly available" to leave authors in suspense. Why not just make the concentration data available now? Also, it is uncertain about the format of single particle data, but is it common practice to archive those data publicly or not?

Response: Both the single particle and bulk data are now available on the University of Miami's repository. DOI link: https://doi.org/10.17604/1427-0558. The link was added to the manuscript.

**R1.C2:** Lines 142-157: It would help readers if you explicitly say up front what part of the text is explaining the method of determining dust mass. It was a bit confusing.

Response: Lines 147-163 describe the procedures for dust collection and extraction. To make this section clearer and easier to follow, we have revised the structure by separating it into two distinct subsections titled "*Dust Mass Concentration Measurement*" and "*Sea Salt Concentration Measurement*"

**R1.C3:** Figure 3: nice visual depiction of results.

Response: We appreciate the reviewer's comment.

**R1.C4:** Line 128-130: The statement *"BACO offers an optimal location for intercepting long-range transported Saharan dust with minimal interference from local anthropogenic emissions due to the prevalent Easterly trade winds"* should be supported with appropriate references.

Response: We have provided appropriate citations (Prospero et al., 2021; Gaston et al., 2024; Zuidema et al., 2019).

**R1.C5:** Line 208-214: The manual SEM/EDX analysis of a relatively small number of particles (40, 21, and 52) raises concerns regarding statistical representativeness and uncertainty. Since manual particle selection can introduce bias, it will strengthen the methodology if the authors clarify how particles were chosen (randomly or selectively) and whether any estimation of analytical or sampling errors was made.

Response: We agree with the reviewer that the relatively small number of manually analyzed particles introduces uncertainty; however, the number analyzed here is sufficient to support qualitative conclusions. Particles were selected randomly across the filter area to minimize sampling bias. Although smaller in number, our dataset is comparable to previous SEM/EDX studies focused on African dust (e.g., Krejci et al., 2005, n = 169 for marine boundary layer dust; Denjean et al., 2015, n = 280 and 170 for dust and background periods, respectively).

We now state in lines 248-260, "*Particles were selected randomly across the filter area without targeting specific particle types or sizes to reduce selection bias. All filter handling was performed in a laminar flow hood, and filters were stored individually in sealed Teflon-taped*

*Petri dishes to avoid any contamination. The number of particles analyzed is reported in Table S2 of the SI."*

*We further state, "To quantify statistical uncertainty, we calculated 95% confidence intervals for the number fraction of each particle class assuming binomial sampling. The major particle types show varying levels of statistical precision. For example, mineral dust is clearly dominant in the SAL (90 ± 9 %) and statistically distinct from mixed dust and sea spray particles, whereas above cloud top and below cloud base, mineral dust and internally mixed dust and sea spray fractions have overlapping confidence intervals, indicating comparable abundance within uncertainty. Thus, while the data robustly supports dust dominance in the SAL, compositional differences among dust and dust mixed with sea spray particle types in the above cloud top and below cloud base should be interpreted qualitatively."*

Table S2: Statistical representation of airborne single-particle measurements showing the 95% confidence intervals for the number fractions of each aerosol particle type.

| | Dust (%) | Dust + Sea Spray (%) | 95% Confidence Interval[a] |
|---|---|---|---|
| SAL | 0.90 | 0.10 | ±0.09 |
| Above cloud top | 0.57 | 0.43 | ±0.21 |
| Below cloud base | 0.42 | 0.58 | ±0.13 |

[a] Confidence intervals are calculated by assuming binomial statistics. For two complementary particle classes (i.e., dust and sea salt), the interval widths are identical because both are constrained by the same sampling variance term.

**R1.C6:** Line 283-287: The explanation of lidar ratio differences between dust (~40 sr) and marine boundary layer aerosols (~20 sr) is unclear and could be misinterpreted. The phrase "a factor of two different with the marine sourced particles being twice as efficient per scattering cross-section compared to dust at backscattering energy" is confusing. Please clarify whether the statement refers to lidar ratio magnitude or backscatter efficiency.

Response: We appreciate this comment and agree that our original phrasing could cause confusion between lidar ratio (extinction to backscatter ratio) and backscatter efficiency. The intent was to emphasize that a lower lidar ratio corresponds to greater backscatter efficiency per

unit extinction, not that marine aerosols scatter more total light than dust. We have revised the text to explicitly describe this physical relationship (lines 328-335) which states:

"*For MAGPIE, the HSRL lidar ratio (LR), the ratio of aerosol extinction ($m^{-1}$) to backscatter ($m^{-1} sr^{-1}$), was approximately 40 sr for dust and 20 sr for marine aerosols. Because LR is inversely related to the particulate 180° backscatter phase function, a lower LR indicates that marine aerosol particles scatter back approximately twice the amount of energy compared to dust if the marine and dust extinctions are the same. This difference in backscatter directly affects the measured LDR. In a mixed aerosol layer with comparable extinction from dust and marine particles, the backscattered signal, on which the LDR is based, is weighed more strongly toward the marine aerosol contribution (that has a lower LDR).*"

**R1.C7:** Line 446-477: The conclusion provides a strong interpretation linking microphysical evidence of dust-sea-spray mixing with the observed low LDR values and their implications for remote sensing. However. The authors may consider strengthening the conclusion by acknowledging additional contributors such as vertical heterogeneity within the MABL (e.g., overlapping marine and dust layers) and potential HSRL retrieval limitations could also contribute to the suppressed depolarization.

Response: As suggested, we have acknowledged the limitations in the revised manuscript. Added text in lines 358-363 reads as below:

"*Nevertheless, we recognize that other factors may also influence the observed reduction in depolarization. Vertical heterogeneity within the MABL, including overlapping layers of marine and dust aerosols, could further convolute the dust depolarization signal. In addition, inherent limitations in HSRL retrievals, such as signal averaging in optically thin layers or reduced sensitivity near the ocean surface may contribute to the apparent underestimation of LDR.*"

**Reviewer #2**

Content Summary

Saharan dust transported over the Atlantic to Barbados undergoes transformations in the marine atmospheric boundary layer (MBL) that are poorly constrained by remote sensing alone. During the August 2023 MAGPIE campaign, the authors combined high-spectral resolution lidar (HSRL) profiles with vertically resolved single-particle chemical and morphological data. They find that while dust in the Saharan Air Layer (SAL) remains externally mixed and yields high lidar linear depolarization ratios (LDR ≈ 0.30), dust descending into the lower MBL becomes internally mixed with sea spray, acquiring more spherical, hygroscopic character that strongly suppresses the depolarization signal (LDR < 0.10) even at high dust loads. The discrepancy between expected and observed LDR is further compounded by differing backscatter efficiencies (lidar ratios) of dust vs marine aerosols, which bias the lidar's sensitivity toward more spherical particles. The authors conclude that neglecting mixing state and morphological evolution can lead to underestimates of surface dust concentrations from depolarization-based retrievals, with implications for satellite retrievals, dust modelling, and air quality analyses.

I consider the topic as interesting and relevant, suitable for ACP. The work adds new detailed data to the pool. The paper written in a clear language and makes a nice reading.

 I have some major points regarding details of the methodology (details further down)

- When making conclusions from the aerosol concentrations and the single particle data for the interpretation of the lidar measurements, the step of including ambient conditions (humidity) is missing. As key statements of this work depend on the comparison, this must be regarded.

- With respect to SEM: there is information missing for the classification schemes and how the shape is calculated. This is relevant for understanding the results and a comparison with similar works. This comparison is largely missing, too; it could however help a lot in assessing the relevance of the present work.

- The number of particles analyzed from the airborne sample is too low to allow for conclusions without a careful characterization of statistical significance.

I encourage the authors to work on the major issues, as I think the work should be published. I hope my comments are helpful.

Response: We sincerely thank Prof. Konrad Kandler for the constructive and thoughtful feedback, which has greatly improved the clarity and rigor in the methodological transparency and the scientific impact of our manuscript. In response, we have made several revisions as suggested:

(1) We have referenced humidity up front in the abstract, and results and discussion. We have also included ambient RH measurements during the campaign (Fig. 1d).

(2) We have expanded the Methods section to include a detailed description of the CCSEM/EDX particle classification and added a comparative discussion with previous studies in the Caribbean to contextualize our findings.

(3) We have addressed concerns regarding statistical representativeness by clarifying the number of particles analyzed, the random selection process, and by estimating 95 % confidence intervals for particle type fractions. These additions demonstrate that, despite the limited number of airborne particles analyzed, the results are statistically comprehensive for qualitative interpretation.

Detailed responses to the reviewer's comments are addressed below:

Details

**R2.C1:** Line 141: As far as I know the BACO tower was constructed to have the top outside the lowermost marine boundary layer. Is it there justified to call the concentrations 'surface'?

Wouldn't it have made sense, if sea salt mixing is a major topic, to compare the top of the tower with measurements at its base or near sea level?

Response: We appreciate the reviewer's comment. The BACO tower is situated on a 30 m bluff with the inlet positioned approximately 17 m above the top of the bluff, a design choice intended to minimize contamination from local surf zone circulation and sea spray generated at the cliff base. Therefore, while the measurements are not taken directly at sea level, they are representative of the near surface MABL and are routinely referred to as "surface" observations in prior Barbados studies (e.g., Zuidema et al., 2019). The tower height effectively reduces local

turbulent influences rather than isolating the sampling from the MABL. We have now clarified this in the manuscript Section 2.1.

**R2.C2:** Line 156-157: How about sulfate and organics, that might come along?

Response: The reviewer is correct. Sulfate and organic aerosol particles were observed during this campaign in the surface samples (see Section S2). We do observe these particles during the dusty period, but most of these particles are of marine origin and were observed during background conditions before the dust intrusion. The absence of these particle types in the airborne samples is likely due to the use of isopore filters with a relatively large pore size (0.8 µm), which may have limited the collection efficiency of finer sulfate and organic rich particles.

We have added a statement acknowledging this limitation associated with the isopore membrane filter in Lines 264-266 which states: "*The sulfate and organic particle types were absent in the airborne samples. This is likely due, in part, to the use of isopore filters with a relatively large pore size (0.8 µm), which may have limited the collection efficiency of finer sulfate and organic rich particles.*"

R2.C3: Line 175: The selection of paper should be rethought. If you want to refer to general CCSEM papers, the first ones were probably in the 80s, like 10.1080/00022470.1983.10465674 and 10.1016/0048-9697(87)90438-4. If you want to refer to the use of CCSEM for mixing state of dust and marine particles, it could include also 10.1126/science.232.4758.1620, 10.1016/S1352-2310(03)00506-5, 10.1029/2005JD005810, 10.5194/acp-18-13429-2018, 10.5194/acp-25-5743-2025, which worked in comparable settings.

 Response: We thank the reviewer for the suggestion. We have added the appropriate references.

**R4.C2:** In many studies, these elements are excluded, if the substrate contains carbon, as the quantification is assumed to have a large error.

Response: We agree with the reviewer that quantification of light elements such as N and O in CCSEM/EDX analysis is subject to greater uncertainty, particularly when using C-coated substrates where signal overlap can occur. In this study, N was not used for quantification, nor did we label it in the EDX spectra of particles. The O signal was used qualitatively to support the identification of organic particles, defined by a combined C + O contribution exceeding 95%.

We have clarified this point in the revised manuscript and acknowledge that absolute quantification of these elements carries higher uncertainty (lines 204-209) which states: "*In contrast, C films are thin and highly transparent to electrons. Although C signals are present in all spectra due to the support film, the C layer is fine-grained and minimally interferes with particle morphology. Moreover, C together with O, serves as a useful qualitative indicator for identifying organic particles, defined by a combined C + O contribution exceeding 95 %. In this study, N was not used for quantification, nor did we label it in the EDX spectra of particles.*"

**R2.C5:** Line 197: When the TEM grids are carbon coated and the film contains carbon, why is only Cu excluded?

Response: The TEM grids consist of a copper mesh coated with a thin (15-25 nm) C support film. Cu, being the grid material, produces strong characteristic X-ray peaks that interfere with particle elemental quantification and were therefore excluded from analysis. In contrast, C films are thin and highly transparent to electrons. Although C signals are present in all spectra due to the support film, the C layer is fine-grained and minimally interferes with particle morphology. Moreover, C together with O, serves as a useful qualitative indicator for identifying organic particle types that exhibit high C and O contributions and can lack other elements. This explanation is added to the text (lines 204-209).

**R2.C6:** Line 202-203: What were the criteria used for that? E.g., how were organics and aged sea spray distinguished (Fig. 3)? For example, if I look at Fig. 5a, the sulfate particle (no. 6) or the aged sea spray (no. 2) have a much higher C signal than the organic shell of no. 5. In fact, C is visible in all spectra.
Some more detailed information is given in supplement, but it does not get clear for example, what were the criteria to separate dust/sea salt mixtures from pure dust or pure sea salt.
This section needs to be a bit more elaborated, so I suggest that a part of the supplement S2 is moved here or at least mentioned and summarized and amended with the missing information.

Response: In the revised manuscript, we have added representative digital color stack plots (as Fig. S3) of representative EDX spectra for major particle type clusters, obtained after running the k-means clustering algorithm, to illustrate the distinguishing elemental features. As suggested by the same reviewer in a later comment, we also added text comparing our results with previous single particle analysis in the Caribbean. The added text in lines 213-234 reads as:

*"Clusters were classified into particle types primarily based on semiquantitative elemental composition obtained from EDX analysis, supported by particle size, morphology, and comparison with prior studies. Mineral dust particles were identified by the presence of aluminosilicate elements (Si, Al, and Fe) characteristic of crustal minerals (Hand et al., 2010; Krueger et al., 2004; Levin et al., 2005; Krejci et al., 2005; Denjean et al., 2015). Fe was detected in ~80 % of mineral dust particles at relative area abundances of 10-15 %. Sea spray particles were characterized by strong Na and Cl peaks, indicative of halite (NaCl) and confirming their marine origin (Bondy et al., 2018). Aged sea spray particles were identified by Cl depletion accompanied by enrichment in S, consistent with heterogeneous reactions that replace Cl with sulfate or nitrate (Ault et al., 2014; Royer et al., 2023, 2025). Mineral dust particles were observed to be both internally mixed with sea spray and externally mixed (Royer et al., 2023, 2025; Kandler et al., 2018; Harrison et al., 2022; Aryasree et al., 2024). These internally mixed dust and sea spray particles exhibited heterogeneous compositions containing both dust-derived (Si, Al, Fe, Mg) and marine-derived (Na, Cl, Mg) components, with Mg potentially originating from both sources. Organic particles were dominated by C and O (>95 %), with minor inorganic elements, and typically appeared as spherical or gel-like structures. Some displayed Mg-rich shells with sea salt cores, consistent with primary marine organics formed via bubble-bursting at the ocean surface (Ault et al., 2013; Gaston et al., 2011; Chin et al., 1998). Sulfate-rich particles exhibited strong sulfur peaks with accompanying C and O signals, indicative of marine secondary aerosols (e.g., ammonium sulfate or bisulfate) and frequently contained an organic fraction (O'Dowd and de Leeuw, 2007; Royer et al., 2023)."*

[Figure]

Figure S3. Digital color stack plots of CCSEM/EDX elemental spectra for representative ground-based particle clusters obtained after running the k-means clustering algorithm. The stacked bars illustrate the characteristic elemental signatures used to differentiate particle classes and the fraction of particles exhibiting each compositional pattern (e.g., Si, Al, Fe, Mg for mineral dust; Na, Cl, Mg for sea salt; S for aged or secondary species; C, O for organics).

**R2.C7:** Line 211: If only 113 particles were analyzed in total, please comment on the statistical significance. How large are the confidence intervals for the percentages given?
Consider this also for the statement made in line 355-357 and 367-369.
I don't doubt the general statement, as it is known from previous studies you refer to that dust over Barbados is not strongly internally mixed, but the low numbers here limit the data applicability for such a statement.

Response: Another reviewer also had similar concerns. Please refer to **R1.C6** for our response.

**R2.C8:** Line 217: Figure 3 seems to be the first figure reference.

Response: We have referenced Section 3.2, instead of figure number, that compares ground-based and airborne single particle analysis to avoid the confusion.

**R2.C9:** Line 238: Why is dry quoted?

Response: The term "dry" is placed in quotation marks to emphasize that the dust above the marine atmospheric boundary layer (MABL) is relatively unmodified and less hygroscopic compared to particles within the humid MABL. We have clarified this in the revised manuscript (line 288-290)

**R2.C10:** Line 256: It seems like the dust AOD precedes the BACO concentrations a bit. Which would make sense regarding the downmixing of the dust from above.

Response: Yes, that is correct. The preceding of coarse AOD relative to the BACO surface concentrations likely reflects both the vertical progression of dust downmixing from the SAL into the MABL and the difference in temporal resolution as the BACO dust data represents ~24 hour integrated samples which may not capture the precise timing of the initial dust intrusion.

**R2.C11:** Line 291-294: "Conducive" seems a bit a misleading word here. Most of the common atmospheric compounds with any hygroscopicity should be expected to be droplets at these humidities. As the soundings were done in the afternoon, it can be expected that the humidity has been higher during other times of the day and, as a result, the particles are on the high branch of the growth vs. humidity hysteresis.

Response: We agree with the reviewer that at RH > 80 %, most hygroscopic aerosol species would already exist on the deliquesced (high) branch of the hygroscopic growth curve. Our intent in using "conducive" was to emphasize that the observed humid conditions further support sustained hygroscopic growth and increase in particle sphericity within the MABL.

We have revised the text to reflect that particles were likely on the high branch of hygroscopic growth during these conditions which reads (lines 389-394):
"*These changes in particle properties caused by hygroscopic growth can further enhance particle backscatter while decreasing the LDR which is visible in the particulate backscatter (Fig. 3a) and LDR (Fig. 3b) measurements below cloud base (~700 m). Thus, under humid*

*MABL conditions, both the LR contrast between dust and marine aerosols and hygroscopicity-driven growth can act together to suppress the observed LDR."*

**R2.C12:** Line 293: Why refer to publications on longwave radiation in the Arctic in this context? Or to a work on Ca and Mg salts (without any direct reference to optical properties)? Remove and replace by suitable ones, if Titos et al. is not deemed to be sufficient.

Response: Because Titos et al. 2016 provides appropriate evidence that hygroscopic growth, i.e., water uptake that increases particle size and scattering efficiency, occurs under high relative humidity, we removed additional citations such as Guo et al., 2019 and Ji et al., 2025.

**R2.C13:** Line 296: Indicate in a, when the soundings were done (e.g., with an arrow or line).

Response: After consideration, we chose not to add arrows or markers because doing so would visually clutter the figure and draw attention away from the key features that we aim to highlight, i.e., the vertical structure of LDR and its evolution during the dust event. The soundings during this period were highly consistent and did not exhibit temporal variability that would meaningfully change the interpretation of Fig. 3a and b. For these reasons, we believe that adding annotations for each sounding would not enhance clarity and may instead distract the reader from the primary message of the figure.

**R2.C14:** Line 296: In Fig. 2a, what is the red feature on Aug 16 at 5 km? Does it coincide with the high humidity?

Response: We have updated Fig. 2a (now Fig. 3 a and b in the revised manuscript) by applying additional masking criteria, specifically filtering data with particulate backscatter values $<10^{-7}$ m$^{-1}$ sr$^{-1}$ and periods influenced by precipitation. The time interval noted by the reviewer corresponds to a precipitation event and is now appropriately filtered in the revised figure.

[Figure]

*Figure 3. HSRL -measurements for (a) particulate backscatter (m⁻¹ sr⁻¹) and (b) particulate linear depolarization ratio (LDR) within 6 km AMSL for August 12 -18, 2023. (c) Vertical profiles of relative humidity (RH, %) up to 6 km AMSL from radiosonde launches at Ragged Point on representative days between August 14 and 18, 2023. In panels (a) and (b), periods with particulate backscatter <10-7 (m⁻¹ sr⁻¹) are masked out. The uncertainty associated with the particulate LDR measurements shown in panel (b) is provided in Fig. S2.*

**R2.C15:** Line 329: The expected LDR should be strongly depending on the humidity, as the sea salt particles would grow by factors of 4 in mass/volume at 80 % and beyond 8 at 95 % RH (vs. dry state), e.g. 10.1038/ncomms15883. As a result, the LDR would be expected to be shifted towards low values at high humidities.

Nevertheless, in the manuscript only the dry sea salt concentrations are compared (line 325-326), and eqns. 7 and 8 in the supplement only seem to take the dry concentration into account (line 164-165). As a result, it seems that this estimate doesn't have much relevance for the humid layers. I suggest including a growth model into the estimate and rethinking the conclusions made (e.g., lines 345-347, lines 365-366).

This also affects the derived statements, e.g. lines 397-399.

Response: We fully agree that relative humidity plays a key role in modulating particle growth and hence the expected LDR, as hygroscopic sea spray particles can undergo substantial size and phase changes at high RH. In fact, our discussion already builds on this physical premise that the humid MABL (>80% RH) provides favorable conditions for hygroscopic growth, leading to more spherical, internally mixed particles and consequently lower depolarization ratios. We have further clarified this link in the revised text (lines 440-443):

"*... Further, these particles likely became even larger under the high relative humidity (>80 %) conditions of the MABL consistent with hygroscopic growth (Zieger et al., 2017). This morphological evolution in internally mixed dust and sea salt particles would explain, in part, the suppressed LDR during the major dust intrusion event (Bi et al., 2022).*"

Regarding the estimation of LDR, we agree that relative humidity and hygroscopic growth are key factors controlling the depolarization signal in the humid MABL. In the revised manuscript, we now explicitly incorporate hygroscopic growth in our estimates. This refined estimate demonstrates that accounting for RH-dependent particle growth further suppresses the estimated LDR, bringing it into closer agreement with the observed values. The updated calculation, described in the SI Text S4 and illustrated in the main text in Figure 6, provides improved closure between measured and predicted depolarization ratios and reinforces our interpretation that internal mixing and hygroscopic growth of internally mixed dust and sea spray particles plays a central role in reducing the lidar depolarization signal under moist MABL conditions. Further, we have added text clarifying theoretical support for low LDR in hydrated internally mixed dust and sea spray particles by Bi et al. (2022), which model light scattering by nonspherical and coated particles, show that when a dust particle becomes coated by a hydrated low refractive index shell (e.g., water, sulfate or sea salt in our case), the spherical shell dominates the backscatter signal and the dust core becomes optically obscured. This mechanism directly supports our interpretation that the high RH in MABL (>80%) produces hydrated internally mixed dust and sea spray particles that suppress LDR despite elevated dust mass. This discussion is added as a new section in the revised manuscript on lines 494-519, which reads:

"*3.3. Accounting for Dust Mixing State and Hygroscopic Growth in Predicting the LDR*

*Prior work by Denjean et al. (2015) showed that externally mixed African dust did not exhibit hygroscopic growth even at high RH (up to 95%), whereas appreciable water uptake occurs*

*primarily when dust is internally mixed with sea spray, a particle type that was prominently observed in our single-particle analysis. Thus, we evaluated how the expected LDR changes when RH-dependent optical weighting is explicitly accounted for by applying a hygroscopic extinction enhancement factor to internally mixed dust and sea spray particles. The detailed discussion of this hygroscopicity dependent calculation is provided in the SI Text S4, and the resulting LDR predictions are shown in Fig. 6. The enhancement factor ($\chi$) represents the marine aerosol extinction enhancement due to the increase in the marine particle cross-sectional area with increasing RH (i.e., hydroscopic growth) (Hänel, 1972, 1976). When this enhancement factor is included, the estimated LDR is further suppressed, consistent with our observations that dust in the moist MABL becomes internally mixed and more spherical when hydrated. This refined estimate improves closure between the measured and predicted depolarization ratios suggesting that hygroscopic growth of internally mixed dust and sea spray particles play a central role in reducing the lidar depolarization signal. Further, simulations of light scattering by nonspherical particles and coated particle systems by Bi et al. (2022) showed that mineral dust particles coated by a hydrated, low refractive index shell (e.g., water, sulfate, or sea salt) can exhibit a strongly suppressed depolarization signal, often approaching values characteristic of spherical particles. This occurs because at high RH the hygroscopic shell grows substantially and dominates the optical response, effectively masking the non-sphericity of the underlying dust core. This coated particle behavior could provide a physical basis for our observations in the humid MABL, where internally mixed dust and sea spray particles observed at RH consistently exceeding 80% produce low LDR values (<0.1) despite high dust mass concentrations and highlights the need to investigate the role of particle composition and mixing state in modulating depolarization signals.*

[Figure]

*Figure 6. Relationship between estimated LDR and dust-to-hygroscopic particle number density ratio as a function of marine aerosol extinction enhancement factor (χ) due to hygroscopic growth. The estimates are based on the observed HSRL-LDR for dry dust particles as 0.3 and LR for dry dust particles as 35 sr and the observed average ratio of cross-sectional area of internally mixed dust and sea spray particles to that of externally mixed dust particles as 2.7, derived from CCSEM/EDX single-particle analysis of surface samples collected at BACO. Cross-sectional areas were calculated using the respective median diameters measured for each particle type.*"

**R2.C16:** Line 338-341: How much contribution would we expect from the particles > 80 μm?

Response: Some particles that large can survive trans-Atlantic transport but are smaller in number concentration compared to the sizes of particles captured by our hi-volume filter sampling. The added text reads as:

"*While previous studies have shown that some particles of this size can survive trans-Atlantic transport (e.g., Betzer et al., 1988; Reid et al., 2003a; Barkley et al., 2021), their number concentrations are expected to be substantially lower than those of the particle sizes efficiently collected by the filter sampling used in this study.*"

**R2.C17:** Line 369-374: this seems to refer to the ground-based measurements, but in line 375 you jump back to the airborne ones. Please order more clearly.

Response: Lines 452-453 now read "*To extend this analysis vertically and examine how particle composition varies with altitude, Fig. 5….*"

**R2.C18:** Line 394-396: How do we learn about the variability of particle composition aloft to compare with? As far as I can see, there is no series of samples available.

Response: The pie charts represent the average particle composition derived from multiple airborne samples collected on different days and at various altitude ranges. To clarify this, we added a supplementary table (Table S1) listing all airborne samples, including their corresponding collection dates, times, and altitude ranges, to document the temporal and vertical coverage of the dataset.

Table S1. Summary of CIRPAS Twin Otter airborne aerosol samples collected during the MAGPIE campaign.

| Date | Time On [UTC] | Time Off [UTC] | Altitude Sample taken [ft] | Cloud Base Height [ft] |
|------|---------------|----------------|----------------------------|------------------------|
| 13AUG23 | 15:02 | 15:30 | 100 | 2300 |
| | 16:35 | 16:50 | 2100 | |
| 14AUG23 | 14:50 | 15:08 | 10000 | 1800 |
| | 15:15 | 15:56 | 500 | |
| 15AUG23 | 15:30 | 15:37 | 3000 | 1000 |
| | 15:40 | 15:46 | 4500 | |
| 16AUG23 | 15:26 | 15:40 | 100 - 300 | 1500 |
| | 16:10 | 16:24 | 7000 - 9000 | |
| 18AUG23 | 14:35 | 15:00 | 100 | 1700 |
| | 16:53 | 17:10 | 4100 | |

**R2.C19:** Line 402: … either … ?

Response: Thank you for pointing out the typo. Removed 'either' from the sentence.

**R2.C20:** Line 403-405: The aspect ratio measured in SEM is representative for SEM conditions, e.g. very dry (vacuum), under which NaCl and other compounds are crystallized. In contrast, in the humid atmosphere we can expect that the sea salt fraction of the particle is in droplet shape.

Therefore, it cannot be expected that the aspect ratio for hygroscopic compounds can be transferred from SEM into the atmosphere. This applies also to line 432.

Response: We thank the reviewer for their thoughtful comments regarding the interpretation of aspect ratios derived from SEM measurements. Several reviewers correctly noted that particle morphology observed under SEM vacuum conditions, especially for hygroscopic particles such as sea salts, does not necessarily reflect the ambient shape or phase state of particles in the humid marine atmospheric boundary layer. Given these valid concerns, and because aspect ratio was not central to our main scientific conclusions, we have removed the aspect ratio analysis and the portion of the corresponding figure from the revised manuscript. Our discussion now focuses on the more physically robust explanations involving hygroscopic growth, internal mixing, and lidar-relevant optical properties under high-RH conditions.

**R2.C21:** Lines 405 is this the average (i.e. daily) size distribution or simply the integral of all particles? In the latter case, was the number of particles analyzed for each sample similar?

Response: The size distributions presented in this section represent the integral of all analyzed particles across each particle class, rather than daily averages. To ensure statistical consistency across samples, we analyzed a comparable number of particles per sample, on average, approximately 1,000 individual particles were analyzed in each sample.

**R2.C22:** Line 407: '… only available from …, …focuses on …' ?

Response: Because significant particle statistics are available from *August 12-17* from our surface measurements, Fig. 5 focuses only on aerosol particles collected at BACO to understand changes in the aerosol size and morphology across different particle types *during the dust event*.

**R2.C23:** Line 417-419: As there are different methods to obtain an aspect ratio from a 2D image, specify the one used, because they yield different results (e.g., doi: 10.1029/2019GL086592 and references therein). In particular, for a square like a cube projection, they do not necessarily come up with 1.

Response: We have addressed this concern earlier (please refer to **R2.C20**); all aspect ratio analysis and discussion have been removed from the manuscript.

**R2.C24:** Line 419: Probably not 'cuboid' here, but 'cubic'. Note that a cube is also aspheric by definition.

Response: This sentence regarding aspect ratio analysis is no longer included in the manuscript.

**R2.C25:** Line 422: If the number after the +/- is the standard deviation, it does not make much sense in the context of a distribution, which is probably rather log-normal than normal. I.e. in this case, the lower end of the standard deviation range would be 0.9, which is an impossible value for the 2D aspect ratio. Either use parameters of a suitable distribution (e.g. 10.1016/j.atmosenv.2007.06.047, 10.1016/j.atmosenv.2015.07.020) or use percentiles.

Response We have addressed this concern earlier (please refer to **R2.C20**); all aspect ratio analysis and discussion have been removed from the manuscript.

**R2.C26:** Line 434: Fig. 5d 'Aged …'

Response: Figure 5d showing the aspect ratio has been removed from the revised manuscript.

**R2.C27:** Line 445: There have been some investigations with CCSEM in and around the Caribbean before (10.5194/acp-22-9663-2022; 10.5194/acp-18-13429-2018; 10.1029/2002JD002935; 10.1098/rsos.231433; 10.5194/acp-5-3331-2005; Roldan, Lizette: Characterization of microphysical properties of Saharan dust aerosols during trans-Atlantic transport. Howard University, 2006; 10.1002/2015GL065693; 10.5194/acp-25-5743-2025). How do these results compare?
E.g. 10.1098/rsos.231433 shows in Fig 5a dust mixing, which is in respect to sea salt similar to the results of the present work (dust mixing on arrival into the boundary layer) but differs for sulfate (already found on the African side). 10.1002/2015GL065693 shows that dust can remain relatively unaltered after trans-Atlantic transport. 10.5194/acp-25-5743-2025 again show considerable mixing at BACO.

Response: We appreciate the reviewer's suggestion to contextualize our CCSEM/EDX results with the previous studies conducted in the Caribbean. In the revised manuscript, we have expanded the description of observed particle types in the Methods section and included a comparison with earlier work in the region highlighting similarities and differences in dust

composition, mixing state, and evidence of aging. For the added text, please refer to lines 213-234 of the revised manuscript or to our response to your comment **R2.C6**.

**R2.C28:** Line 449-450: Combining data from Fig. 3 (assuming that the data is representative) and from Fig. 5: if the dust becomes gradually more mixed during the downward-mixing in the boundary layer, why do we see less dust/sea salt mixtures at BACO compared to the flight measurements?

Response: We appreciate the reviewer's comment and agree that the lower fraction of internally mixed dust and sea spray particles observed at BACO, compared to the aircraft samples, is not fully resolved by our current dataset. A likely explanation is that the aircraft sampled below cloud base and above cloud top, where turbulent mixing, entrainment, and cloud processing are strongest and where conditions are more favorable for the formation of internally mixed dust and sea spray particles. In contrast, the near surface samples at BACO are continually influenced by freshly emitted sea spray and may not receive the same degree of downward transport of dust or conditions required to promote internal mixing. We also note that the 95% confidence intervals calculated for the airborne samples are relatively wide for both the above cloud top and below cloud base samples due to limited particle counts.

**R2.C29:** Like 452-455: Check if the statement can be kept up.
Response: We removed the word spherical from the sentence.

**R2.C30:** 'Peak' refers to conditions at BACO?

Response: Changed to: "Despite peak dust loading **at BACO** (AOD ~0.75; surface dust ~120 µg/m³)…."

**R2.C31:** Supplement, S2 reference to SEM images is not correct (now 5a or missing?)

Response: We thank the reviewer for catching this error, and we replaced "Fig. 3c" with "Fig. 4a" in the revised manuscript.

**R2.C32:** Fig S2: Commonly kernel density estimators are used to smooth a histogram. But these curves are not smooth. Why? What estimator was used? Or is that a size distribution with a density on y? Please check.

Response: We thank the reviewer for catching this. The figure was mistakenly labeled as showing a kernel density estimate. In fact, the curves represent binned probability density distributions derived from normalized particle counts (i.e., probability per unit size). Specifically, we used MATLAB's built-in histcounts function normalized by the total count and bin width to obtain a probability density function, which preserves the true bin to bin variability in the measured size distribution. We did not apply additional kernel smoothing. We have corrected the Figure label in the revised version to read "Probability density distribution".

**R2.C33:** In general: check the capitalization of the labels in the plots. E.g. 5a, image no. 1: capital at the start. No 2.: capital at the second word. No. 5: All words capitalized.

Response: Thank you for pointing it out. We have revised the Figure labels.

**Reviewer #3**

This paper presents measurements of dust and sea salt over Barbados acquired during August 2023 and describes the impact of these aerosols on the ground-based HSRL measurements of depolarization and backscatter. The major focus of the paper is in assessing how these aerosols mix to reduce the lidar linear depolarization ratio (LDR) measured within the marine boundary layer (MBL), thereby frustrating efforts to use the LDR as an indicator of dust. This is an important topic as lidar measurements of LDR are often used to quantify dust amounts and evaluate model predictions of dust transport. The authors use tower and airborne measurements of particle size and composition to show that large amounts of dust were present even when the LDR was low (<0.1). The authors conclude that using LDR will often result in underestimates of surface dust concentration and argue that in situ measurements must be combined with such lidar measurements to correctly determine dust concentration.

The topic is suitable for ACP. The paper is generally easy to read and publication is recommended after the authors address the comments below.

We appreciate the reviewer's positive assessment and recommendation for publication. Detailed responses to the reviewer's comments are addressed below.

**R3.C1:** Line 34 (and elsewhere). "linear depolarization ratio" It's not clear here (and elsewhere) whether this means volume (or total) linear depolarization ratio or particle depolarization ratio. (see description in the Burton et al. 2015 reference). From the values provided, it appears to be particle depolarization, but this should be clearly indicated here and elsewhere.

Response: The linear depolarization ratio presented in this study corresponds to the particulate linear depolarization ratio. We have clarified this in the revised manuscript.

**R3.C2:** Line 35 The wavelength of the lidar measurement should be indicated (532 nm).

Response: Thank you. We added the wavelength of the lidar measurement.

**R3.C3:** Section 2.1 Did the authors ever consult the MPI Raman lidar measurement images that are available on-line at https://barbados.mpimet.mpg.de/? These show extensive measurements of aerosol backscatter and depolarization over Barbados that also tend to support the HSRL

measurements presented in the paper. These images exist for several years and include August 2023.

Response: We thank the reviewer for the comment. These MPI Raman lidar data indeed offer valuable historical context for aerosol backscatter and depolarization variability over Barbados. Our analysis, however, focuses on the coordinated multi-platform observations collected during the MAGPIE field campaign at the Ragged Point site, where the University of Wisconsin SSEC HSRL system provides calibrated retrievals of extinction, backscatter, and depolarization with high signal-to-noise ratio (SNR) during both day and night, including at very low altitudes near the ocean surface, capabilities that are essential for interpreting MABL aerosol structure in this study.

To acknowledge the broader observational context, we have added reference to the MPI Raman lidar record in Section 2.6 (Weinzierl et al., 2017; Groß et al., 2015; Stevens et al., 2016). While not directly integrated into our analysis, this long-term dataset complements our campaign focused work by providing multi-year evidence of the vertical distribution of aerosol layers over Barbados.

Added text in the Section 2.6 (lines 278-281) where we discuss the HSRL observation reads: "*Long-term Raman lidar measurements from the Max Planck Institute (MPI) in Barbados (Weinzierl et al., 2017; Groß et al., 2015; Stevens et al., 2016) provides historical context for aerosol backscatter and depolarization over the island and show structures consistent with the HSRL observations presented here.*"

**R3.C4:** Line 212. This section discusses airborne particulate samples for single particle analysis. If the CTO inlet has a cut point of 3.5 mm, how were samples as large as 25 mm sampled and manually analyzed?

Response: We thank the reviewer for the thoughtful question. The nominal inlet cut point for the CTO inlet, corresponding to the 50% collection efficiency diameter, is approximately 3.5 µm aerodynamic diameter. This means that while collection efficiency decreases above this size, larger particles can still be sampled at lower efficiencies, particularly under high aerosol loading conditions such as during the dust intrusion period. Thus, although particles up to ~25 µm were

occasionally observed, these represent the relative upper tail of the inlet transmission under strong dust conditions rather than the typical sampled size range.

**R3.C5:** Line 214. Can the authors be more specific about how the first maximum in the relative humidity profile was assigned to be the CBH? How large did this maximum in RH have to be? Were these CBH values compared with those that can be readily determined from the HSRL measurements?

Response: During the initial sounding at each sampling station, vertical profiles of air temperature and dew point temperature were monitored as they gradually converged with increasing altitude. The CBH was identified at the altitude where the air temperature equaled the dew point temperature, indicating 100% RH and the onset of condensation. This ascent or descent profiling strategy was performed during each flight to establish sampling levels and capture the vertical moisture structure critical for identifying the cloud base. We have clarified this in SI section S1.

**R3.C6:** Line 225. Readers looking for the details of the SSEC HSRL are supposed to consult the Razenkov and Eloranta references. What are the uncertainties in the HSRL measurements of aerosol backscatter, depolarization, and lidar ratio? I could not find these in the Razenkov reference. I also wonder whether this reference is still relevant for measurements acquired 15 years after the thesis was written (i.e. has the instrument and analyses remained the same during this period?) The Eloranta reference discusses the design and construction of the NCAR airborne HSRL; note that this reference is not readily available at my institution. I also wonder if this reference provides such uncertainty estimates and if so, whether they apply to the ground based lidar in the same way to the airborne lidar. Given that the HSRL measurements of LDR (and to a lesser extent lidar ratio) are a major topic of this paper, there should be at least a brief discussion of the uncertainties in these measurements; such discussion is absent from the paper. How large are the uncertainties in the volume and particulate depolarization and lidar ratio?

Response: The cited references accurately describe the foundational HSRL system design, and we have now supplemented these with a more recent reference relevant to the in-field HSRL configuration and data usage during the ONR PISTON cruise and NASA CAMP$^2$EX flight campaigns (Reid et al., 2025). In addition, as suggested by the reviewer, we have incorporated a description of the systematic uncertainties associated with the HSRL measurements in the

revised manuscript. Please refer to lines 358-363 or **R1.C7** for the added text in the revised manuscript.

Further, we have also included the figure for systematic LDR uncertainty standard deviation (as Fig. S2 in the SI) associated with the HSRL scan for particulate LDR (shown in Fig. 2a in main text).

[Figure]

Figure S2. The systematic uncertainty standard deviation associated with the HSRL scan for particulate linear depolarization ratio (LDR; shown in Fig. 3b in main text) within 6 km above MSL for August 12 -18, 2023.

**R3.C7:** Line 228. When operating pointing vertically, how close to the surface are profiles of aerosol backscatter and depolarization obtained?

Response: While our HSRL measurements were made up to ~50 m above mean sea level, the HSRL data remain reliable up to ~135 m. When the HSRL transmits laser pulses, the photon detectors are saturated for several nano seconds due to photon scattering inside the instrument. At 135 m, the photon detectors are no longer saturated. Accordingly, we have updated Fig. 1c to show the LDR and particulate backscatter at 135 m instead of 105 m, which provides the most reliable comparison for this study.

**R3.C8:** Figure 1c shows that the largest LDR (105 m) occurred on 08-23-23, yet the dust concentration at the top of the tower (Fig. 1a) was very small (negligible?) Why? This seems to indicate that there can be substantial differences between 50 m (top of tower above sea level) and the lowest lidar measurement height (105 m). Were there measurements made at/near the base of the tower or closer to sea level to study the vertical variations close to the surface?

Response: The elevated LDR on 23 August is associated with a short period of offshore flow that advected an aerosol plume with higher depolarization characteristics from island sources. This feature is not indicative of a substantial or systematic difference between the BACO tower height and the lowest HSRL measurement level. Rather, it reflects the difference in temporal resolution between the instruments. The tower-based dust sampler integrates over a 24-hour period, which smooths out short-duration events, whereas the HSRL detects changes at minute-level temporal resolution. As such, brief inflow episodes that produce sharp but transient increases in LDR would not appear in the daily integrated filter measurements.

**R3.C9:** Figure 1. It would be interesting to see wind speed also during this period to see if/how the depolarization, backscatter, and lidar ratio varied with wind speed and also the amount of sea spray.

Response: We appreciate the reviewer's suggestion. Wind speed and RH data have now been added to Figure 1. As discussed in the forthcoming Reid et al. (BAMS, in preparation) paper, COAMPS simulations indicate that wind speeds were elevated during the dust intrusion period, consistent with the high wind speed observed during the event (Fig. 1d). We did not observe a correlation between wind speed and sea spray concentrations, at least at the daily sampling resolution of our filter-based measurements.

[Figure]

Figure 1. Time series plots for (a) dust and sea salt mass concentrations measured from the top of the BACO tower, (b) AERONET total column and fine mode fraction AODs (at 500 nm), (c) HSRL- particulate linear depolarization ratio (LDR) and particulate backscatter at 532 nm, averaged over six hours, and (d) meteorological measurements (RH and wind speed) during the MAGPIE 2023 campaign. The red dashed box represents the major dust event observed during the campaign.

**R3.C10:** Line 284. "…depolarization measurement responds to the 180-degree backscatter efficiency of the particulates (lidar ratio)." This is why there needs to be a better description of the LDR that is referred to. The particulate depolarization depends on the total (volume) depolarization as well as the particulate scattering ratio. The sentence currently is confusing since it mentions (lidar ratio) at the end of the sentence. I could easily see how a reader not familiar with lidar (and HSRL) can become confused.

Response: We have clarified in the revised manuscript that the LDR presented in this study corresponds to the particulate LDR.

**R3.C11:** Line 288. Following on the last comment, "…the measured depolarization will be weighted lower due to the backscatter efficiency difference between the aerosol (i.e., lidar ratio)". Lower than what? Do the author mean instead (or also) "…the measured backscatter will be weighted lower due to the backscatter efficiency difference between the aerosol (i.e., lidar ratio)".

Response: Another reviewer also had similar comments. Please refer to **R1.C6** for our response.

**R3.C12:** Line 309. Maybe this will be discussed later in the paper, but what is the basis of the statement "…due to the predominance of larger but less backscattering mineral dust particles".

If the RH is higher in the MBL, and the particles are hygroscopic, wouldn't the particles in the MBL be larger?

Response: We thank the reviewer for this observation. Yes, this statement is supported later in the manuscript (Fig. 3c), where we show that externally mixed mineral dust dominates the upper layer, while the lower MBL contains more internally mixed dust and sea spray particles. Regarding the reviewer's second point, we agree and note that this hypothesis was already present in our discussion that the higher RH in the MABL promotes hygroscopic growth. Consistent with this, we also observed that internally mixed dust and sea spray exhibit larger sizes compared to externally mixed dust or fresh sea spray particles.

Please refer to lines 326-335 in the revised manuscript or **R1.C6** for the revised text.

**R3.C13:** Figure 3. It would be interesting to see profiles of the lidar ratio (532 nm) and backscatter color ratio (ratio of backscatter at 532 to 1064 nm) for this case also. These are two important aerosol parameters that can also help in interpreting the aerosol vertical distribution.

Response: We appreciate the reviewer's helpful suggestion. The current HSRL system deployed during MAGPIE includes a 1 µm channel, however, this channel is presently not fully calibrated for scientifically reliable retrievals and therefore cannot be used to derive a robust backscatter color ratio or 532/1064 nm lidar ratio profile for this case. Recent developments within the SSEC HSRL team have, for the first time, implemented a fully calibrated 1 µm HSRL system, which is expected to be deployed in future field campaigns. Once available, color ratio and multi-wavelength lidar ratio measurements will provide valuable additional constraints on aerosol vertical distribution and mixing state. We have noted this as an avenue for future work in the revised discussion.

**R3.C14:** Line 363. "underestimated" should be "overestimated".

Response: We appreciate the reviewer's careful reading. The statement is correct as written. The observed LDR is underestimated by the calculated expected LDR based on dust and sea salt mass concentrations and lidar ratios. To clarify this point, we have revised the sentence (line 369 to read:

"*The calculated LDR was approximately a factor of two higher than what was observed during the peak dust event.*"

**R3.C15:** Line 379. I don't understand why the transition layer is described as "indifferentiable". The LDR clearly shows this layer as different from the SAL and MBL.

Response: Thank you for pointing this out. The reviewer is correct, and we have removed the term 'indifferentiable' from the sentence in the revised manuscript.

**R3.C16:** Figure 5. What do the black lines in 5b and 5c represent?

Response: Figure 5 in earlier submission now corresponds to Figure 4 in the revised manuscript. The black line in Fig. 4b corresponds to the dust mass concentration and that in Fig. 4c is total number of analyzed particles. We have clarified this in the Figure caption.

**R3.C17:** Figure 5c. This seems to show only sea spray and not aged sea spray. Is that correct?

Response: Figure 5 in earlier submission now corresponds to Figure 4 in the revised manuscript. Figure 4c includes both fresh sea spray and aged sea spray particles. However, because the number fraction of aged sea spray particles was relatively small compared to fresh sea spray and the colors used had close contrast, they may not have been visually distinct at first glance.

**R3.C18:** Figure 5d. Are these measurements made at low RH? If so, wouldn't the aspect ratios be very different at ambient RH?

Response: We have addressed this concern earlier (please refer to **R2.C20**); all aspect ratio analysis and discussion have been removed from the manuscript.

**R3.C19:** Line 449. Should be "internally" `

Response: We have corrected the typo. Thank you.

Lines 467-477 This paragraph expresses the desire/need for combining lidar data with in-situ single particle analysis to improve the interpretation of lidar data in dust regions. While this may be true for the best interpretation of such data, this can't be done routinely and globally on a continuous basis; that is why remote sensing techniques are pursued. The authors seem to indicate that there are no other alternatives and all remote sensing techniques are doomed to significantly underestimate dust impacts near the surface. Have the authors considered whether more advanced remote sensing measurements could provide additional data to help improve the interpretation of lidar data? For example, measurements of backscatter, extinction, depolarization at additional wavelengths? As a suggestion, the reviewers may want to examine the backscatter color ratio (or Angstrom exponent) using various wavelengths (ex. 355-532 nm) in such situations, especially when acquired by such HSRL systems. Examination of such data, in conjunction with depolarization data at multiple wavelengths suggests that, while the depolarization near the surface may be low suggesting that dust concentrations are low, the backscatter color ratio has similar values as observed in the SAL region and which are also different from values in other MBL regions where dust values are low, suggesting that backscatter color ratio may be an indicator of dust. The point is that the authors should not prematurely dismiss remote sensing techniques for providing accurate estimates of dust loading simply because of the limitations of the lidar measurements studied here.

Response: We appreciate the reviewer's thoughtful perspective and fully agree that advanced multi-wavelength remote sensing provides a promising pathway for improving the interpretation of dust in the lower MABL. Our intention was to highlight that under certain conditions, single wavelength depolarization alone may be insufficient without additional information on aerosol mixing state. As the reviewer notes, multi-wavelength measurements, particularly backscatter color ratio and depolarization at additional wavelengths (e.g., 355, 532 and 1064 nm), offer additional constraints that can help differentiate dust from hydrated marine particles even when LDR is low. This is entirely consistent with ongoing efforts by the SSEC HSRL team. Recent upgrades to the SSEC HSRL have produced the first fully calibrated 1-micron HSRL system, and the goal of the SSEC HSRL team is to implement this capability in future field deployments. Such measurements would allow color ratio-based indicators of dust to be evaluated alongside depolarization, thereby providing a more robust remote sensing framework independent of in-situ sampling. At the same time, multi-wavelength lidar products still require independent validation to ensure physical consistency. This study provides precisely that reference point: vertically resolved, single particle chemical and size measurements that clarify how dust optical properties evolve as it mixes with sea spray in the MABL. Without such independent constraints, it would be difficult to determine whether discrepancies in lidar-derived dust estimates arise from aerosol transformation, retrieval assumptions, or instrument limitations. Thus, rather than dismissing remote sensing approaches, our results highlight the importance of combining advanced lidar measurements validated with targeted in-situ observations to achieve accurate dust retrievals near the surface.

We have added text in the revised Discussion acknowledging this important point and emphasizing that multi-wavelength HSRL observations represent a key next step for improving near surface dust detection using remote sensing techniques like HSRL.

See below the added text in the Conclusion section (lines 565-579):
"*While our results demonstrate that single wavelength depolarization can underestimate near surface dust under humid, mixed aerosol conditions, we emphasize that more advanced remote sensing approaches can mitigate these limitations. Multi-wavelength HSRL observations, including backscatter at 532, and 1064 nm and corresponding color ratio and depolarization metrics, provide additional degrees of freedom for discriminating dust from hydrated marine*

*aerosol particles. In fact, recent upgrades by the SSEC HSRL team have produced the first calibrated 1064 nm HSRL system, that is aimed at being deployed in future studies. These multi-spectral measurements would enable color ratio signatures characteristic of dust to be detected even when LDR is low, thereby providing a remote sensing pathway to constrain surface dust loading. Validating these multi-spectral retrievals requires independent constraints on aerosol composition and morphology. The vertically resolved single particle measurements presented here provide validation of how dust properties change as they mix with sea spray. Thus, rather than diminishing the utility of lidar, our results highlight the importance of integrating advanced multi-wavelength lidar products with targeted in-situ observations to improve the accuracy of surface dust estimates in marine environments.”*

**Reviewer # 4**

This study investigated the evolution characteristics of physical and chemical properties of Saharan dust during its interaction with sea salt aerosols during long-range transport, and carefully analyzes the vertical distribution characteristics of the aging process of dust aerosols in the marine atmospheric environment, which is of great importance for understanding the radiative properties of dust aging. The results also highlight the importance of integrating vertically resolved lidar data with in-situ single-particle analysis and surface aerosol mass concentrations to improve the interpretation of lidar observations in dust-affected regions. The manuscript's presentation of linguistic logic is clear and rigorous, and its overall writing quality is good. Nevertheless, there are some minor issues in the manuscript that require further revision and clarification. Only when the following issues have been revised or clarified is it recommended for publication:

We appreciate the reviewer's positive assessment and recommendation for publication. Detail responses to the reviewer's comments are addressed below.

**R4.C1:** The manuscript contains a large number of abbreviations, and it is recommended to include a list of abbreviations.
Response: We added a list of abbreviations as Appendix 1 in SI.

**R4.C2:** Section 2.2 and Figure 1a: How is the dust mass concentration derived?

Response: Please refer to our response to previous reviewer's comment **R1.C2**.

**R4.C3:** Section 2.2: The method for calculating sea salt concentration is based on the assumption that all $Na^+$ originates from sea salt aerosols. However, dust aerosols contain a certain amount of sodium salts. Although the proportion of $Na^+$ in dust is small, when the dust concentration is high, the $Na^+$ contribution from dust may be difficult to ignore. Therefore, it is recommended to recalculate the sea salt concentration after deducting the $Na^+$ from dust aerosols.

Response: As mentioned in the manuscript (Line 161-163), halite is not a major constituent of Saharan dust, and previous studies have shown that its contribution rarely exceeds 3% by weight. The presence of sodium in our samples is therefore more consistent with sea spray influence rather than a mineral dust source. Further, if Saharan dust had appreciable Na content, we would

expect a concurrent increase in sea spray concentrations during the dust intrusion event, however, such a trend was not observed in our measurements.

**R4.C4:** Line 247: The part after "0.03" lacks a period.

Response. Added a period.

**R4.C5:** It is recommended to add meteorological data on Figure 1, such as temperature, relative humidity (RH), wind speed/direction. In particular, RH can assist in understanding the role that meteorological conditions within dust plumes played in altering the physicochemical properties of aged dust (dust + sea salt) with high hygroscopicity.

Response: We thank the reviewer for this valuable suggestion. We have added meteorological parameters (RH and wind speed) to Figure 1d to provide additional context. We also note that the radiosonde sounding data presented in Figure 3c effectively represents the vertical RH structure and captures the surface level humidity conditions during the dust intrusion period.

**R4.C6:** The interaction between dust aerosols and sea salt over the ocean has long been studied. For instance, about 20 years ago, Zhang et al. investigated the interaction between dust aerosols originating from the Asian continent in East Asia and sea salt aerosols in the northwestern Pacific Ocean. It is recommended that the authors compare this study with previous research to highlight the innovations of this paper. (References: Zhang, D. Z., et al., Geophys. Res. Lett. 2001, 28 (18), 3613-3616; Zhang, D. Z., et al., Mixture state and size of Asian dust particles collected at southwestern Japan in spring 2000. J. Phys. Chem. A 2003, 108 (D24); Zhang, D. Z. and Iwasaka, Y., Size change of Asian dust particles caused by sea salt interaction: Measurements in southwestern Japan. Geophys. Res. Lett. 2004, 31 (15); Zhang, D. Z., et al., Coarse and accumulation mode particles associated with Asian dust in southwestern Japan. Atmos. Environ. 2006, 40 (7), 1205-1215.)

Response: We thank the reviewer for pointing out these relevant studies, which we have now incorporated into the revised manuscript and referenced in the discussion of particle types.

**R4.C7:** Figure 5b shows that when dust concentration is relatively high, the fraction of sulfate particles is also high. Why? Are these sulfate particles derived from anthropogenic emissions or natural sources? Is there a possibility that these sulfate particles originate from dust aerosols?

Recent studies have shown that fresh dust aerosols also contain sulfate (Li, W. et al., A Review of Water-Soluble Ions in Natural Dust Particles Over East Asia: Abundance, Spatial Distributions, and Implications. ACS ES&T Air 2025, 2 (8), 13791393).

Response: We agree with the reviewer that the elevated sulfate fraction observed during periods of high dust loading may arise from multiple sources. While sulfate in the MABL is generally of marine origin (e.g., from dimethyl sulfide oxidation), it is also plausible that a portion of the sulfate was co-transported with dust from continental or anthropogenic sources. As the reviewer correctly notes, fresh dust can contain sulfate internally mixed with mineral phases. This explanation is consistent with recent findings by Gaston et al. (2024), who reported a gradual increase in non-sea salt sulfate in decades of sample collected at the BACO site associated with anthropogenic influence from Africa. We have added this clarification to the revised manuscript.

References:

[revised manuscript text omitted]